# Phosphatases in Mitosis: Roles and Regulation

**DOI:** 10.3390/biom9020055

**Published:** 2019-02-07

**Authors:** Margarida Moura, Carlos Conde

**Affiliations:** 1IBMC—Instituto de Biologia Molecular e Celular, Universidade do Porto, 4200-135 Porto, Portugal; Margarida.moura@ibmc.up.pt; 2i3S—Instituto de Investigação e Inovação em Saúde da Universidade do Porto, 4200-135 Porto, Portugal; 3Programa Doutoral em Biologia Molecular e Celular (MCbiology), Instituto de Ciências Biomédicas Abel Salazar (ICBAS), Universidade do Porto, 4050-313 Porto, Portugal

**Keywords:** mitosis, phosphatases, cell division cycle 25 (CDC25), protein phosphatase 1 (PP1), protein phosphatase 2A (PP2A), kinetochores, microtubules, chromosomes

## Abstract

Mitosis requires extensive rearrangement of cellular architecture and of subcellular structures so that replicated chromosomes can bind correctly to spindle microtubules and segregate towards opposite poles. This process originates two new daughter nuclei with equal genetic content and relies on highly-dynamic and tightly regulated phosphorylation of numerous cell cycle proteins. A burst in protein phosphorylation orchestrated by several conserved kinases occurs as cells go into and progress through mitosis. The opposing dephosphorylation events are catalyzed by a small set of protein phosphatases, whose importance for the accuracy of mitosis is becoming increasingly appreciated. This review will focus on the established and emerging roles of mitotic phosphatases, describe their structural and biochemical properties, and discuss recent advances in understanding the regulation of phosphatase activity and function.

## 1. Introduction

Protein phosphorylation is the most profuse post-translational modification in eukaryotes. It plays a critical role in controlling protein function, activity, and molecular interactions. The phosphorylation state of a given protein is established by the balance of activities between protein kinases and protein phosphatases. Kinases catalyze the transfer of a phosphate group to tyrosine (Tyr), threonine (Thr), or serine (Ser) residues, while protein phosphatases are responsible for the counteracting hydrolytic removal of the phosphate moiety. Phosphoproteomic analyses revealed that protein phosphorylation occurs primarily at Ser and Thr residues (83% and 15% of sites, respectively), with Tyr phosphorylation accounting for only up to 2% of the total phosphoproteome [1]. The importance and regulatory role of protein kinases is well established, whereas protein phosphatases were long regarded as passive housekeeping enzymes. However, it is now evident that phosphatases are also subjected to stringent control and assume equally critical roles in the regulation of protein function and signaling responses. The human genome contains 189 known and predicted protein phosphatase genes [2]. Their evolutionary history can be traced back to a core catalytic domain that evolved through a series of gene duplication events and by adopting the use of regulatory subunits and/or fusion with novel functional modules or domains [2]. Thus, contrasting with kinases, whose catalytic domains are commonly from a single protein structural fold [3], protein phosphatase activity is associated with numerous different protein folds and catalytic mechanisms [4,5]. Based on substrate specificity, protein phosphatases are typically classified as serine/threonine phosphatases, tyrosine phosphatases, or as dual-specificity phosphatases [6].

### 1.1. Serine/Threonine Phosphatases

Serine/threonine phosphatases are distributed through three broad families in accordance to their primary structure and catalytic mechanism: the phosphoprotein phosphatase (PPP), the protein phosphatase metal-dependent (PPM), and the haloacid dehydrogenases (HAD) families [5]. The most abundant serine/threonine phosphatases in eukaryotic cells are protein phosphatase 1 (PP1) and protein phosphatase 2A (PP2A) [6]. Given their vital importance for cell cycle progression, PP1 and PP2A will be discussed in greater detail. These phosphatases are members of the PPP family, which also includes PP2B, PP4, PP5, PP6, and PP7. All PPPs are phosphomonoesterases (EC3.1.3) and share a similar core catalytic domain, with two metal ions at the active site. This bimetallic center, typically a Zn::Fe in native enzymes or a Mn::Mn pair in recombinant PPPs, is bridged by a water molecule that acts as a nucleophile in the hydrolysis of phosphorylated Ser (pSer) and pThr substrates [6,7,8,9]. The core catalytic domains of all PPPs exhibit modest substrate specificity when assayed in vitro. Unlike most kinases, PPPs fail to recognize a consensus sequence surrounding or adjacent to the phosphorylated residue [10,11,12,13,14]. Tight substrate selectivity is, however, achieved in a cellular context through association with specific regulatory subunits, which also ensure the control of the phosphatase subcellular localization and modulate overall activity [4,6]. Despite overall similarities, PP1 and PP2A catalytic subunits show some, albeit limited, intrinsic substrate specificity in vitro due to structural variations in their catalytic sites. Protein phosphatase 2A, for instance, is inhibited by subnanomolar concentrations of okadaic acid, due to hydrophobic residues in its catalytic cleft that bind tightly to okadaic acid, whereas PP1 inhibition requires 100-fold higher concentrations [15,16]. Furthermore, isolated catalytic subunits of PP1 and PP2A show distinct preferences for different subunits of phosphorylase b kinase [17]. However, there is compelling evidence that PP1 and PP2A do not exist as free catalytic subunits in cells but operate as multimeric holoenzyme complexes composed of the catalytic subunit and by one or more regulatory and scaffolding subunits that control the phosphatase specificity and activity under physiological settings.

#### 1.1.1. Protein Phosphatase 1

A PP1 holoenzyme contains one of four isoforms of the PP1 catalytic subunit—PP1α, PP1β, PP1γ1, or PP1γ2 (alternative splicing variant of PP1γ only expressed in testis) and one or two of the numerous regulatory subunits [18,19,20,21]. Catalytic subunits of PP1 show extreme phylogenetic and functional conservation. This is illustrated by the ability of any human PP1 isoform to rescue PP1/Glc7 loss in budding yeast [22]. Biochemically, PP1α, PP1β, PP1γ1, or PP1γ2 are identical enzymes [5,6,23] with extensive functional overlap, although some specific functions have been attributed to PP1γ2 in testis and to PP1β in some tissues [20,21,24]. Protein phosphatase 1 isoforms adopt a compact α/β fold, with a β sandwich wedged between two α-helical domains [25,26]. The catalytic site resides at the intersection of three potential substrate/adaptor binding regions: the hydrophobic, acidic, and C-terminal grooves. These are shallow and converge at the catalytic center, forming a Y-shaped surface feature (Figure 1A) [5].

More than 200 regulatory subunits of PP1 have been described. These are generically designated as PP1-interacting proteins (PIPs) and its diversity provides the cell with a wide array of different PP1 holoenzymes to meet specificity and regulatory requirements [27,28]. PIPs are mostly structurally unrelated proteins whose interaction with PP1 is frequently mediated by short linear motifs (SLiMs) that bind to the surface grooves of PP1 (Figure 1B) [29,30,31,32]. Several distinct PP1-binding motifs have now been characterized, enabling a huge combinatorial potential to generate unique PP1 holoenzymes with specific properties [6,27,28]. The most frequent and best characterized PP1-binding SLiM is the RVxF-motif. This motif conforms to the consensus sequence [K/R]x_(0–1)_[V/I]x[F/W] and is present in nearly 70% of identified PIPs [33,34]. Structural insights revealed that the RVxF-motif docks into a hydrophobic pocket on the PP1 surface through contacts that are primarily mediated by the hydrophobic side-chains of valine (Val) and phenylalanine (Phe) [35]. Additional examples of PP1-binding SLiMs include the SILK, MyPhoNE, and SpiDoC motifs [27,28]. The binding of PIPs to PP1 can also occur through highly structured segments, such as leucine-rich repeats or ankyrin repeats, respectively present on SDS22 and MYPT1 regulatory subunits [36,37,38]. Different PP1-binding motifs associate with different regions of the phosphatase surface. Therefore, a given PP1:PIP interaction might be mediated by several PP1-binding motifs to provide a higher affinity and enhance the complex stability. Importantly, although the same PP1 surface can establish contact with different PIPs, the interaction occurs in a highly specific manner due to the unique combination of PP1-binding motifs in each PIP.

Crystallography studies show that binding to PIPs does not impose a discernible conformational change on the PP1 catalytic domain. Instead, most PIPs limit PP1 activity towards specific substrates through a combination of functional domains involved in PP1-anchoring, PP1-inhibition, substrate-recruitment, and/or subcellular-targeting [6,27,28]. Therefore, controlled dephosphorylation can be achieved by a multitude of mechanisms. Some PIPs, such as spinophilin or protein phosphatase 1 nuclear-targeting subunit (PNUTS), bind to the PP1 C-terminal groove and sterically hinder the access to substrates recruited through this region [39,40]. Nuclear inhibitor of protein phosphatase 1 (NIPP1), on the other hand, defines PP1 substrate selectivity through a polybasic stretch in its PP1-anchoring domain, which dramatically changes the electrostatic charge distribution of the NIPP1:PP1 surface, and by a N-terminus forkhead-associated (FHA) domain that mediates the recruitment of specific substrates [41]. Other PIPs target PP1 to specific subcellular compartments through domains that may also be involved in substrate recruitment. This raises the local concentration of PP1 and enables the dephosphorylation of resident substrates. The regulatory subunit inhibitor-2 (I-2) binds to the acidic and hydrophobic grooves of PP1. This positions the IDoHA motif in contact with the catalytic center, obstructing its access to substrates and causing the release of metals required for catalysis [42]. Other PIPs, such as inhibitor-1 (I-1), CPI-17, and MYPT1, associate with PP1 in a phospho-dependent manner and may inhibit its activity by posing as pseudosubstrates or act as anchoring subunits [26,43,44,45]. Several heterotrimeric PP1 holoenzymes have been identified where PP1 associates with one targeting- or substrate-specifying PIP and one inhibitory PIP. The I-1, I-2, inhibitor-3 (I-3), and CPI-17 are examples of inhibitory PIPs that are able to interact with various PP1-PIPs dimers. The crystallographic structure of PP1 bound to spinophilin, and to I-2 has revealed major structural rearrangements in both regulatory subunits to assemble the heterotrimeric complex [46,47]. The RVxF motif of spinophilin has a higher affinity for the hydrophobic pocket on PP1 and displaces the RVxF motif of I-2, which remains, however, anchored to PP1 through a SILK motif. The IDoHA motif of I-2 dislodges the ∂-helix of spinophilin. These rearrangements enhance the flexibility of the holoenzyme, which is thought to increase its ability to bind substrates or provide new protein interaction sites that were previously concealed and thereby mediate access to additional targets. In summary, amongst the vast repertoire of identified PIPs, there are subcellular/substrate-specifiers, inhibitors, and substrates that also act as inhibitory regulators of PP1. Their combinatorial association with each of the PP1 catalytic subunits provides the cell with a multitude of PP1 holoenzymes with unique structural and biochemical properties to enable the spatiotemporally controlled dephosphorylation of specific targets under physiological conditions.

#### 1.1.2. Protein Phosphatase 2A

The PP2A catalytic subunit has two isoforms (PP2ACα and PP2ACβ) that share 97% of their sequence identity with each other [48,49]. Like other PPPs, PP2A catalytic subunits operate as members of discrete holoenzyme complexes to achieve substrate specificity and controlled activity in cells. PP2A holoenzymes are predominantly heterotrimers consisting of a catalytic subunit (C) bound to a scaffold subunit (A), which in turn recruits a variety of mutually exclusive regulatory subunits (B) with distinct substrate and subcellular specificities. The human genome encodes two scaffolding subunits that share 87% of their sequence identity with each other (PP2AAα and PP2Aaβ isoforms), and 16 regulatory subunits grouped into four families according to their primary sequence and structure: B55 (PR55/B), B56 (PR61/B′), B72 (PR72/B′′), and Striatin (PR93/B′′′) [50,51].

The PP2A scaffold subunit contains 15 tandem helical HEAT (huntingtin-elongation-A subunit-TOR, target of rapamycin) repeats assembled into an elongated, horseshoe-shaped structure [52] (Figure 2A). PP2A(C) specifically binds to the conserved rim of HEAT repeats 11–15 to form the heterodimeric core enzyme [15]. Although the PP2A catalytic subunit shares extensive sequence similarity with other PPPs, the majority of the latter fail to associate with PP2AAα or PP2AAβ because the interaction is mediated by specific amino acids that have been replaced by nonconserved residues in PP1, PP2B, PP5, and PP7 [5]. Roughly one-third of PP2A catalytic subunits are present in mammalian cells in the form of heterodimers with a scaffolding subunit [53]. Whether this PP2A heterodimeric complex is physiologically relevant or merely represents a readily available intermediate towards holoenzyme assembly remains unclear.

The association of a regulatory B subunit to the core enzyme provides substrate specificity as well as spatially and temporally controlled functions [54]. Structural analysis revealed that the B56 subunit extensively interacts with both the scaffold and the catalytic subunit (Figure 2A). The B56 subunit contains eight HEAT-like repeats and is structurally similar to the scaffold subunit. The convex surface of B56 binds to HEAT repeats 2–8 of the scaffold subunit, whereas the concave surface is tilted toward the active site pocket of the catalytic subunit to define the substrate-recognizing surface (Figure 2A) [55,56]. The B55 subunit on the other hand adopts a WD40 fold. It comprises a seven-bladed β propeller and a protruding β-hairpin handle that respectively bind to the HEAT repeats 3–7 and 1–2 of the scaffold subunit [57]. Contrasting with the B56 subunit, B55 makes few contacts with the catalytic subunit and the interaction with substrates is mediated by the acidic central groove of the β propeller (Figure 2B). Thus, notwithstanding evident structural differences and their specific interactions with the scaffold subunit, both B55 and B56 generate PP2A holoenzymes in which the substrate-binding site is located on the top face of the regulatory subunit at a position that is proximal to the active site of the catalytic domain [5] (Figure 2A,B). Different B subunits generate different substrate-recognition interfaces. The PP2A core enzyme is only able to efficiently dephosphorylate the microtubule-binding protein, Tau, when in complex with the B55 subunit [57]. Two lysine-rich sequences in Tau are predicted to bind to the acidic groove on the top face of the β propeller. The residues on B55 required for the interaction have been mapped, but further studies will be required to establish the precise Tau sequences that contact with B55 [57]. Additional critical insight into the B55 substrate recognition mechanism was recently provided through a combination of phosphoproteome screens and kinetic modelling. The strategy allowed the global identification of B55 mitotic substrates, most of which conform to the cyclin-dependent kinase 1 (CDK1) phosphorylation signature [58]. This specificity of B55 is provided by a bipartite positively charged polybasic motif located upstream and downstream of the CDK1 site that acts as a binding determinant for the acidic surface of B55 [58]. Notably, substrates that are dephosphorylated more rapidly by PP2A-B55 are generally more basic than substrates that are dephosphorylated at slower rates [58]. This strongly suggests that the cooperative electrostatic interactions between the negatively charged residues in B55 and the positively charged amino acids on the polybasic motif control the dephosphorylation kinetics [58,59]. Moreover, PP2A-B55 exhibits a preference towards pThr over pSer residues that is not determined by electrostatic interactions with B55, as it has also been observed in other PP2A holoenzymes [60,61,62,63]. This preference seems to reflect instead an inherent property of the PP2A catalytic subunit. It is possible that the additional methyl side group of threonine could affect the binding affinity or even the catalytic reaction [64].

Recent studies shed new light on the mechanisms by which PP2A-B56 holoenzymes engage with some of its substrates. The B56 subfamily in humans comprises five isoforms (α, β, γ, δ, and ε) with no discernible differences in their substrate binding pockets [51]. These subunits recognize SLiMs conforming to the [LMFI]xx[ILV]xE sequence, henceforward referred to as the LxxIxE motif [65,66,67,68]. Structural analysis of B56γ interaction with the kinetochore protein, BubR1, or with the nuclear scaffolding protein, Repo-Man, revealed that the LxxIxE motif binds to a conserved basic pocket between the third and the fourth HEAT repeats on the concave surface of B56γ [66,67] (Figure 2C,D). Peptide phage display of intrinsically disordered regions of the human proteome identified 126 ligands of B56 and confirmed substantial specificity overlap between PP2A-B56α and PP2A-B56γ holoenzymes as expected from the highly conserved LxxIxE binding surface in all B56 isoforms [68]. Nevertheless, B56 isoforms were recently shown to bind differentially to LxxIxE motifs on mitotic proteins [69]. This apparent selectivity is proposed to allow B56α and B56γ to localize at discrete subcellular compartments and control separate mitotic processes. Thus, although functional redundancy has been attributed to B56 isoforms, some sub-functionalization appears to occur under certain circumstances. The molecular features underlying B56 isoform-specific interactions during mitosis are discussed in Section 2.4.

### 1.2. Tyrosine Phosphatases

Although recurrently underrepresented in global phosphoproteomes, tyrosine phosphorylation plays a critical role in numerous signal transduction pathways that underlie a broad spectrum of fundamental physiological processes. Protein tyrosine phosphatases (PTPs) are defined by the active-site signature motif HCX5R, in which the cysteine residue acts as a nucleophile essential for catalysis [70,71]. Protein tyrosine phosphatases are categorized as classical tyrosine-specific phosphatases (CTSPs) or as dual-specificity phosphatases (DUSPs) [4,72,73]. Contrasting with PPP catalytic subunits, PTPs are selective enzymes that exhibit striking substrate specificity in a cellular context. Hence, instead of occurring in vivo in complex with a multitude of regulatory subunits, PTPs comprise a range of structurally distinct catalytic units that are encoded by nearly 100 genes in the human genome [70,74,75]. The use of alternative promoters, alternative splicing, and post-translational modifications introduce further structural diversity [70]. The catalytic domain of classical PTPs (CTSPs) comprises approximately 280 residues and is defined by the HCX5R motif that functions as a phosphate-binding loop at the active site and by a mobile general acid/base loop (WPD) [4]. Despite relatively large sequence variations in the X5 segment, the P-loop conformation is strictly conserved and ensures that the cysteine and arginine residues remain in close proximity to efficiently remove the phosphate group from the substrate tyrosine. The sulphur atom of the cysteine thiolate serves as a nucleophile and attacks the phosphorus of the phosphotyrosyl target, whereas arginine contributes to substrate binding and stabilizes the cysteine-phosphate intermediate [76,77]. This intermediate is resolved by an activated water molecule aided by an aspartate in the WPD loop [76,77]. Dual-specificity phosphatases are the largest sub-group of PTPs with approximately 70 members [70]. These phosphatases also encompass a typical HCX5R signature motif in their active site, but have smaller catalytic domains and are more structurally diverse than the CTSPs. Notably, the active site pocket of DUSPs is wider and shallower, which allows them to accommodate pSer and pThr residues as well as pTyr residues [78]. Dual-specificity phosphatases share the same catalytic mechanism as the CTSPs [78]. Although DUSPs do not possess a WPD loop, a conserved aspartate residue in the corresponding position is proposed to act as a general acid/base during catalysis [75]. With the notable exception of the inhibitory phosphorylation of CDK1 on Tyr15, reversible tyrosine phosphorylation was long thought to play a minor role in mitosis. However, SFKs, ABL1, EGFR, and PKM2 are examples of PTPs whose inhibition was shown to compromise mitotic accuracy [79,80,81,82]. Furthermore, the master mitotic regulator, polo-like kinase 1 (PLK1), was recently shown to be phosphorylated on Tyr217 during mitosis [83]. This phosphorylation occurs within the kinase activation segment and is proposed to repress the activity of selected pools of PLK1 during cell division [83]. Hence, available new evidence suggests that mitotic pTyr signaling may be more prevalent than previously appreciated. Nevertheless, a comprehensive and mechanistic knowledge of mitotic events controlled by tyrosine phosphorylation remains limited and the only PTPs with well-established essential roles in mitosis are the dual-specificity phosphatases, cell division cycle 14 (CDC14) and cell division cycle 25 (CDC25) [84,85,86]. Phosphatases of the CDC25 family have evolutionarily conserved functions in controlling key transitions between cell cycle phases. CDC14, on the other hand, is a key mitotic exit phosphatase in budding yeast, but its relevance for mitotic exit in metazoans remains ill-defined. Instead, a growing body of evidence indicates that mitotic exit in animal cells relies primarily on regulatory networks orchestrated by PP1 and PP2A phosphatases [86].

#### CDC25 Phosphatases

The human genome encodes three CDC25 phosphatases: CDC25A, CDC25B, and CDC25C [87,88,89]. The three isoforms are ~60% identical in their C-terminal catalytic domain, but diverge considerably in their N-terminal regulatory region [90,91]. All three CDC25 phosphatases act at various points of the cell cycle to control the activity of specific CDK-Cyclin subpopulations [92,93,94]. Not surprisingly, CDC25 phosphatases are often overexpressed in cancer cells [90]. Structural characterization of CDC25A and CDC25B catalytic domains revealed a near identical folding that is clearly distinct from that of any PTP [95,96]. The small catalytic domains adopt a rhodanese-like structure with a central five-stranded parallel β-sheet sandwiched by three α-helices from below and two α-helices from above. The active sites containing the HCX5R signature are flat and shallow and lack auxiliary loops or obvious features for mediating substrate recognition. Rather, this seems to rely on three hotspot residues (R488, R492, and Y497 on CDC25B) that form a substrate docking site remotely located from the catalytic cysteine [91,97,98]. In line with that, the primary sequence surrounding the site of dephosphorylation does not seem to contribute significantly for substrate recognition [99]. For instance, CDC25A and CDC25B phosphatases exhibit a clear preference towards phosphorylated CDK2 (on Thr14 and Tyr15) in complex with Cyclin A (CDK2-pThr14pTyr15-CyclinA) over artificial substrates or phosphorylated peptides that harbor the same target sequence [91]. Both catalytic domain constructs and respective full-length phosphatases exhibit the same affinity for CDK2-pThr14pTyr15-CyclinA, indicating that all the elements required for substrate recognition are in the catalytic domain. Interestingly, CDC25C is significantly less active towards CDK2-pThr14pTyr15-CyclinA, but highly efficient in dephosphorylating CDK1-pThr14pTyr15-CyclinB [100]. Such differential reactivity is attributed to differences in the C-terminal tail of each CDC25 phosphatase. The C-terminus corresponds to the least conserved region of the catalytic domain and remains undefined in crystal structures, presumably due to its intrinsic flexibility or disordered nature. Nevertheless, two arginine residues present in the C-terminal tail of CDC25A and CDC25B, but absent from CDC25C were shown to dramatically enhance the affinity for CDK2-pThr14pTyr15-CyclinA [91]. The catalytic mechanism of CDC25 phosphatases is very similar to the well-established mechanism of PTPs [91,101,102]. The cysteine in the active site forms a transient covalent phospho-cysteine. Departure of the leaving group and formation of the intermediate are facilitated by protonation of the oxyanion by aspartate acting as the catalytic acid. In a second chemical step, water serves as a nucleophile in the hydrolysis of the phospho-enzyme intermediate assisted by the aspartate, which now acts as a base to activate the water molecule. Extensive experimental evidence confirmed this general mechanism for CDC25 phosphatases, but the identity of the catalytic acid remains to be unambiguously established [91,103,104,105].

## 2. Roles of Protein Phosphatases in Mitosis

The unidirectional transitions between the phases of the cell cycle is governed by reversible phosphorylation and proteolysis. The highest occupancy of phosphorylation sites is observed in mitosis. Over 32,000 protein phosphorylation/dephosphorylation events occur as cells go into, progress through, and exit from mitosis [106,107,108,109]. Such an extensive network of dynamic phosphorylations underlies the physical events that distribute one copy of the genome to each daughter cell and their subsequent resumption to an interphase state. Mitotic entry in animal cells involves dramatic cellular and structural reorganization, including cell rounding, disassembly of the nuclear envelope, condensation of chromatin into tightly packed chromosomes, and formation of the microtubule-based mitotic spindle. These events occur in concert with regulatory and surveillance pathways to enable the attachment of chromosomes to microtubules from opposite spindle poles and are primarily driven by elevated activity of the CDK1-Cyclin B complex. This attachment configuration allows sister chromatids to be pulled towards opposite poles during mitotic exit, hence providing the same genetic content to progeny cells. To safeguard the accuracy of chromatid partitioning, cells only commit to irreversibly exit mitosis when sister kinetochores of each chromosome attain proper amphitelic attachments. Chromosome segregation and decondensation, cytokinesis, and reassembly of interphase cell structures are the major events defining mitotic exit, all of which require inactivation of CDK1 as well as spatially and temporally confined removal of mitotic phosphorylations to occur. These are respectively driven by the degradation of Cyclin B and controlled activity of protein phosphatases.

### 2.1. Mitotic Entry: CDC25 Phosphatases Set up the Commitment

Activation of CDK1-Cyclin B has long been established as the trigger that drives the transition into mitosis. The activity of CDK1 throughout the cell cycle is regulated by the quantity and subcellular localization of Cyclin B and by the phosphorylation status of CDK1 itself. Expression of Cyclin B increases during the G2 phase of the cell cycle, thus allowing formation of the CDK1-Cyclin B complex before mitotic entry [110]. The activity of CDK1-Cyclin B is however repressed by inhibitory phosphorylations on CDK1 Thr14 and Tyr15 catalyzed by WEE1 and MYT1 kinases [111,112]. These residues are in the CDK1 catalytic domain and their phosphorylation induces conformational changes in the ATP-binding loop that render the kinase inactive. Dephosphorylation of pThr14 and pTyr15 catalyzed by CDC25 dual-specificity phosphatases is considered the rate-limiting step in the transition from G2 into mitosis (Figure 3A). In human cells, all three CDC25 isoforms contribute for mitotic entry. The CDC25B isoform is responsible for the initial activation of CDK1-Cyclin B at centrosomes during late G2 [113,114,115]. The kinase is subsequently translocated into the nucleus, where it becomes completely activated by CDC25C at the onset of mitosis [116]. The CDC25A phosphatase is proposed to activate CDK1-Cyclin B complexes that trigger chromosome condensation [113,117,118,119]. Partially active CDK1-Cyclin B directly stimulates CDC25B and CDC25C activities and promotes the inhibition of WEE1 and MYT1 kinases, hence establishing both a positive and a double-negative feedback loop that drives swift and sustained CDK1 activation and the irreversible transition to mitosis (Figure 3A) [120,121,122,123,124].

In addition to CDK1-Cyclin B, multiple other mechanisms contribute to regulate CDC25 phosphatases (Figure 3B). These include changes in CDC25 protein levels and subcellular localization, inhibitory phosphorylations, and binding to regulatory proteins. The CDC25A isoform is predominantly expressed in G1 and subjected to rapid turnover during interphase as a result of SCFβTRCP ubiquitin-mediated degradation [125]. The phosphatase becomes, however, stabilized at the G2/M transition upon CDK1-Cyclin B-mediated phosphorylation on Ser17 and Ser115 [118,126], thus contributing to the phosphatase pool required to fully activate CDK1-Cyclin B. The levels of CDC25A decrease rapidly at the end of mitosis due to APC/C-Cdh1-mediated targeting for proteasomal degradation [125]. The levels of CDC25B begin to accumulate during mid S-phase, peak at mitotic entry, and decrease as a result of SCFβTRCP-dependent degradation at the metaphase-anaphase transition [114,127,128,129,130]. Contrasting with the other isoforms, CDC25C protein levels remain constant throughout the cell cycle [92]. However, the phosphatase is kept inactive during interphase due to Ser216 phosphorylation. This phosphorylation can be catalyzed by a number of kinases, including Chk1, Chk2, C-Tak, CaMKII, and PKA [131,132,133,134,135,136], and generates a binding site for the 14-3-3 family of proteins [137,138,139]. The interaction of CDC25C with 14-3-3 masks a nuclear localization signal, resulting in cytoplasmic sequestration of the phosphatase and consequently in the inhibition of mitotic entry [92,119,140]. Moreover, studies with Xenopus oocytes indicate that pSer216 might also directly suppress CDC25C catalytic activity independently of 14-3-3 binding [141]. Similarly, phosphorylation of CDC25B on the equivalent residue (Ser323) by p38 MAPK or pEG3 promotes binding to 14-3-3 and thereby cytoplasmic retention of the phosphatase during interphase. Interestingly, 14-3-3 also binds to CDC25A following phosphorylation of the latter on Thr505 by CHK1. This, however, does not affect the phosphatase nuclear localization but rather impairs its interaction with CDK1-Cyclin B to control proper timing of mitotic onset [142].

Mitotic kinases have also been shown to operate as active regulators of CDC25 phosphatases (Figure 3B). Centrosome-localized Aurora A phosphorylates CDC25B on Ser353 during prophase. Although the molecular underpinnings remain elusive, this phosphorylation correlates with the relocalization of Cyclin B to the nucleus and CDK1 activation during mitotic entry [143,144]. Aurora A is also implicated in promoting timely activation of CDC25C, though indirectly through PLK1 [145]. While the requirement of PLK1 activity for mitotic entry upon the G2 DNA-damage checkpoint is well established, its relevance during unperturbed cell cycles has long remained controversial [146,147,148,149,150,151]. Recent studies have now demonstrated that complete inhibition of PLK1 delays mitotic entry even in the absence of DNA-damage events. PLK1 is rapidly activated in late G2 shortly before CDK1-Cyclin B and is dependent on Aurora A and CDK1-Cyclin A activities [152,153,154]. Polo-like kinase 1 subsequently phosphorylates the CDC25C N-terminal region on multiple sites, promoting its nuclear import [155,156] and possibly directly stimulating the phosphatase catalytic activity [146,149]. Therefore, Aurora A- and PLK1-mediated control of CDC25 phosphatases represent critical events in the molecular circuitry that ensures robust CDK1-Cyclin B activation and mitotic commitment (Figure 3B).

Data from Xenopus oocytes revealed that prompt CDC25C activation further requires the activity of PP1 phosphatase (Figure 3B). During mitotic entry, CDK2 activity promotes the dissociation of 14-3-3 protein from CDC25C, exposing the inhibitory phosphorylation on Ser287 (corresponding to Ser216 in the human CDC25C orthologue) for PP1-mediated removal [157]. This enables the initial activation of CDC25C and a consequent increment in CDK1-Cyclin B activity. CDK1-Cyclin B subsequently phosphorylates CDC25C on Ser285 and enhances the recruitment of PP1 to CDC25C, hence accelerating S287 dephosphorylation and thereby warranting prompt and sustained phosphatase activation [141].

Along with increased CDK1-Cyclin B activity, the inhibition of PP2A-B55 phosphatase is essential for mitotic compliance (Figure 3A). B55-type regulatory subunits confer specificity of PP2A towards CDK1 phosphosites, which suggests that PP2A-B55 holoenzymes might represent a critical opposing activity to CDK1-Cyclin B [57]. Accordingly, PP2A-B55 mediated removal of numerous CDK1-generated phosphorylations was shown to be critical for mitotic exit [58,63,158,159,160,161,162,163,164]. In order to prevent premature dephosphorylation of CDK1-Cyclin B substrates and ensure mitotic commitment, PP2A-B55 activity is repressed by the Ser/Thr kinase Greatwall (GWL) at the G2/M transition (Figure 3A). Two key studies using Xenopus egg extracts uncovered the underlying biochemical pathway. GWL phosphorylates Arpp19 and ENSA, two small and unstructured proteins of the Endosulfine family. Phosphorylation occurs on specific serine residues (Ser62 in Arpp19 and Ser67 in ENSA) within a highly conserved FDSpGDY motif and allows Arpp19 and ENSA to bind PP2A-B55 in a highly specific manner, causing its near-complete inhibition [165,166]. This mechanism was also shown to be essential for mitotic entry in flies, starfish oocytes, and in human cultured cells [167,168,169,170,171,172]. A notable exception was observed in mouse embryonic fibroblasts, where the control of PP2A-B55 by GWL seems to be dispensable for the G2/M transition, but nevertheless required after nuclear envelope breakdown to prevent mitotic collapse [173,174]. As expected, and by opposition to PP2A-B55, GWL activity is undetectable throughout interphase, abruptly increases at the G2/M transition, and remains elevated until anaphase onset [169,171,175]. Activation of GWL is triggered by CDK1-Cyclin B-mediated phosphorylation of two conserved residues located in the kinase presumptive activation loop. This primes GWL to intramolecularly autophosphorylate its C-terminal tail and stabilize the kinase in its active conformation [175,176,177]. In frame of the G2/M signaling circuitry, the inhibition of PP2A–B55 orchestrated by GWL has been proposed to increase the levels of activating phosphorylation events on CDC25 phosphatases and inhibitory phosphorylation events on WEE1 and MYT1 [86,178] (Figure 3A).

### 2.2. Sister Chromatid Cohesion: PP2A-B56 Holds Them Together

Mitotic entry is accompanied by the condensation of DNA into chromosomes, each comprised of two sister-chromatids held together from DNA replication until anaphase onset by proteinaceous ring-like cohesin complexes [179,180] (Figure 4A). These are composed by heterodimers of SMC1 and SMC3 bound to the α-Kleisin subunit, SCC1/MCD1/RAD21, forming a closed tripartite structure that encircles the replicated DNA [181]. The Kleisin subsequently binds to SCC3/SA, which when associated with the heterodimer, WAP1-PDS5, forms a subcomplex that regulates the maintenance of cohesion on chromatin [182,183] (Figure 4B). During prophase, cohesin is displaced from chromosome arms by WAP1, possibly through disruption of the SMC3-SCC1 interface (Figure 4A,B) [179,184,185,186,187,188]. Prior to mitotic entry, however, cohesin rings are insensitive to WAP1 as a result of SMC3 acetylation by ECO1/2 acetyltransferases [189,190,191,192,193]. Acetylation of SMC3 during DNA replication promotes the recruitment of Sororin, which antagonizes WAP1 by impairing its binding to PDS5 [194,195]. Notably, loss of protection against WAP1 during early mitosis does not involve SMC3 deacetylation. It is instead elicited through extensive phosphorylation of Sororin and SCC3/SA2 by Aurora B, CDK1, and PLK1 [196,197,198,199,200,201,202,203]. Aurora B- and CDK1-dependent phosphorylation of Sororin causes its dissociation from PDS5, hence consenting WAP1 binding to PDS5 (Figure 4A,B) [199,201]. Moreover, WAP1-mediated removal of cohesin further requires phosphorylation of SCC3/SA2 by PLK1. The underlying mechanism remains, however, unknown. Loss of cohesion from chromosome arms in prophase seems to occur exclusively in metazoan cells, where is thought to facilitate sister DNA decatenation and allow timely cohesin reloading in the subsequent cell cycle [204,205,206,207]. Cohesion must, however, persist at centromeres until the kinetochores of all chromosomes are correctly attached to microtubules of opposite spindle poles. Centromeric cohesin facilitates this process by promoting the back-to-back arrangement of sister kinetochores and grants resistance to microtubules pulling forces so that separation can only occur upon mitotic checkpoint silencing [208,209,210,211].

Centromeres are protected against WAP1-dependent cohesin removal by the activity of SGO1-associated PP2A-B56 [199,212,213,214,215]. Recruitment of PP2A-B56 to centromeres of mammalian cells is mediated both by SGO1 and SGO2 [216,217,218]. Although SGO2 is required for cohesion protection during meiosis, it seems to be dispensable for this function in mitotic cells, where instead SGO1 assumes primary relevance [212,213,215,216,217,219,220,221,222,223]. Structural analysis of an N-terminal fragment of human SGO1 bound to PP2A-B56γ reveals a bipartite interaction between SGO1 and the regulatory and catalytic subunits of the PP2A-B56γ holoenzyme [218]. SGO1 N-terminus forms a parallel coiled-coil homodimer whose N- and C-terminal ends bind to PP2AC and B56γ, respectively. Each binding interface contains a hydrophobic core as well as peripheral hydrogen bonds and salt bridges. Binding of SGO1 to PP2A-B56γ does not seem to impose significant alterations in PP2AC structure or catalytic activity. In a cellular context, this interaction likely places PP2A-B56γ in close proximity to Sororin and SCC3/SA2 to counteract Aurora B, CDK1, and PLK1 destabilizing phosphorylations and thereby prevent the dissociation of centromeric cohesin complexes (Figure 4B). Several lines of evidence concur in support of this model: binding of PP2A-B56γ to SGO1 is required for protection of centromeric cohesion [215,218], SGO1 physically interacts with cohesin complexes during mitosis through SCC1–SCC3/SA2 heterodimers, and an unphosphorylatable version of Sororin prevents chromatid separation that would normally follow SGO1 depletion [199].

### 2.3. To Build a Spindle: A PP1, PP2A, PP4, and PP6 Groundwork

Accurate segregation of chromosomes relies on the formation of a bipolar spindle structure [224]. This requires the maturation and separation of duplicated centrosomes to originate two independent microtubule-organizing centers correctly positioned at opposite poles of the cell. Centrosome maturation occurs when cells prepare to enter mitosis and results from a dramatic expansion in the organized matrix of pericentriolar material (PCM) surrounding mother centrioles and a concomitant increase in its microtubule nucleation capacity [225,226,227]. Expanded PCM serves as a catalytic scaffold for γ-tubulin ring complexes (γ-TuRC) to mediate microtubule nucleation [228,229] and its assembly involves the recruitment of numerous PCM proteins orchestrated by PLK1 and Aurora A kinases [227,230]. The Ser/Thr phosphatase PP4 is predominantly enriched at mitotic centrosomes of metazoan cells and its activity was shown to be required for γ-tubulin accumulation and microtubule nucleation in *Drosophila* and *Caenorhabditis elegans* embryos, as well as in human cultured cells. The mechanism by which PP4 promotes the maturation of centrosomes remains unknown, but the phosphatase appears to be necessary for proper centrosomal localization of PLK1 [231] and activation of the the Aurora A-CEP192 complex [232]. Conversely, disruption of PP4 in MEFs has no impact on γ-tubulin levels, but leads to unstable contacts between PCM and microtubules minus-ends [233].

Regulation of microtubule dynamics at the centrosomes and spindle poles relies in part on the activity of the microtubule-severing Katanin complex [234,235], whose centrosomal localization is mediated by the interaction of its p60 subunit with the Dynein-motor regulator NDEL1 [236]. Binding of p60 Katanin to NDEL1 increases during mitotic entry as a result of NDEL1 phosphorylation at Ser198, Thr219, and Ser231 by CDK1-Cyclin B [236]. These phosphosites are directly antagonized by PP4 phosphatase, which limits the accumulation of Katanin activity at centrosomes and thereby promotes the stabilization of microtubule connections [233]. This is important to enable organized outgrowing of microtubules into a functional mitotic spindle, a process that additionally requires discrete activities of PP2A and PP6 phosphatases. Interestingly, in *C. elegans*, PP4 directly dephosphorylates the Katanin catalytic subunit homologue MEI-1 to promote microtubule-severing activity in meiosis [237,238].

Plus-end growth of microtubules emanating from centrosomes relies on the activity of the microtubule-polymerase ch-TOG/XMAP215/MSPS [239], whose mechanism was recently uncovered in *Xenopus* egg extracts [240]. The C-terminus of XMAP215 directly interacts with γ-tubulin whereas the N-terminus TOG domains bind to soluble α/β-tubulin dimers to endorse their assembly onto γ-TuRCs [240]. The number of microtubules that grow out from centrosomes is limited by the activity of the microtubule depolymerizing kinesin MCAK/XKCM1/KLP-7 [239,241]. In *C. elegans* embryos, the regulatory subunit, PR72/RSA-1, targets PP2A to centrosomes through a direct interaction with the PCM-associated RSA-2 protein. Centrosomal PP2A-PR72/RSA-1 promotes microtubule outgrowth and spindle stability by controlling KLP-7 and the spindle assembly factor, TPX2/TPXL-1. The PP2A-PR72/RSA-1 complex restricts the levels of KLP7 through an unknown mechanism and directly mediates the accumulation of TPX2/TPXL-1 at centrosomes, thus increasing the rate of microtubules’ nucleation as well as the length of centrosomal microtubules [242]. No obvious RSA-2 orthologues have been identified so far and the requirement of PP2A-PR72 for spindle formation in other metazoans remains to be demonstrated. In fact, TPX2 displays limited centrosomal localization in vertebrate cells, accumulating preferentially on spindle microtubules [243,244], where it plays a critical role in recruiting and promoting Aurora A activity required for chromatin-driven spindle assembly [245,246,247]. Following nuclear envelope breakdown, the RanGTP that is produced by the chromatin-associated GEF RCC1 relieves TPX2 from inhibitory interactions with Importin-α/β [246,248,249]. This licenses TPX2 to bind and allosterically activate Aurora A [243,250,251,252]. The TPX2-Aurora A complex oligomerizes and binds to the RHAMM-γ-TuRC complex, which becomes competent for microtubule nucleation after Aurora A-mediated phosphorylation of the γ-TuRC adaptor subunit, NEDD1 [253,254,255]. Interestingly, microtubule-associated TPX2-Aurora A is more active in the purlieu of spindle poles. This is owed to the spatially controlled activity of PP6 phosphatase and is critical for the stability of the assembling spindle [256]. In addition to allosteric regulation by TPX2 binding, dimerization-mediated T-loop autophosphorylation (Thr288) further enhances Aurora A activation [257,258,259]. Moreover, the interaction with TPX2 protects the activating phosphorylation from the antagonizing action of several phosphatases, including PP1 and PP2A [250,251,252,260,261]. The Ser/Thr phosphatase, PP6, is, however, able to recognize and dephosphorylate the T-loop of the TPX2-complexed form of Aurora A and its depletion was shown to impair proper spindle assembly [256,262]. Although PP6 is uniformly distributed through the cytoplasm of mitotic cells, its activity towards TPX2-Aurora A is inhibited in the vicinity of centrosomes by action of PLK1 [263]. Following CDK1-dependent priming, PLK1 binds to and phosphorylates the PP6-regulatory subunit, PP6R2, at multiple sites [263]. These phosphorylations repress PP6 activity possibly by preventing its interaction with substrates. Because PLK1 is particularly enriched at centrosomes, PP6-mediated dephosphorylation of TPX2-Aurora A complexes located on proximal microtubules is repressed, thus ensuring maximal Aurora A activation at spindle poles and moderate activation on distal spindle microtubules [259,264,265]. This enhances Aurora A-catalyzed phosphorylation of transforming acidic coiled-coil (TACC) proteins at spindle poles, which enables TACC-ch-TOG complexes to efficiently bind to and stabilize the minus-ends of nascent microtubules and, in this way, promote growth of microtubules from the centrosome [241,266,267,268,269,270,271,272,273,274,275]. Hence, coordinated activities of PLK1, PP6, and Aurora A favor the centrosomes as dominant sites of spindle assembly in mitosis.

Setting two independent microtubule-organizing centers at opposite poles of a mitotic cell requires the separation of duplicated centrosomes, which typically occurs before nuclear envelope breakdown in organisms that undergo an open mitosis [276]. Centrosome splitting during late G2 involves the dissolution of a fibrous proteinaceous linker that holds the duplicated centrosomes together, a process elicited by an increase in NEK2A kinase activity and controlled by PP1 phosphatase [277,278,279,280] (Figure 5). The primary target of NEK2A is C-Nap1, which is docked to the proximal ends of parental centrioles through the centriolar protein, CEP135, and serves as a platform for the assembly of Rootletin/CEP68 fibers [281,282,283,284,285]. Data from STED microscopy and 3D-STORM analysis suggest that Rootletin/CEP68 filaments coming from each centriole are entwined into a web-like structure that probably underpins the flexible nature of the centrosomal linker [286]. Multisite phosphorylation of C-NAP1 by NEK2A abrogates C-NAP1 interaction with CEP135, hence causing its displacement from centrosomes [287]. This concurs with NEK2A-mediated phosphorylation of Rootletin and of associated proteins to ultimately disrupt the linker structure and enable centrosome disjunction (Figure 5) [281,287,288,289,290]. The kinase activity of NEK2A is tightly regulated to prevent premature centrosome separation, whose occurrence was shown to result in aberrant nuclear positioning and incorrect spindle orientation [291,292,293]. Activation of NEK2A occurs through homodimerization-induced autophosphorylation of the activation loop [294,295,296]. The kinase is also phosphorylated in its non-catalytic domain by sterile-20 like kinase MST2, which significantly enhances its localization at centrosomes [297]. However, NEK2A activity towards C-NAP1, and possibly other substrates, is kept at residual levels up until late G2 by action of PP1γ, which binds to the NEK2A-MST2-hSAV1 complex through an RVxF motif on the NEK2A C-terminus [298,299,300]. Within the complex, PP1γ antagonizes NEK2A phosphorylations on C-NAP1, thus maintaining the linker integrity and, consequently, the centrosomes tethered [299]. Whether PP1γ or PP1α dephosphorylates the NEK2A T-loop to directly inactivate the kinase remains unclear [300,301]. Loss of centrosome cohesion is triggered by the activation of PLK1, which subsequently phosphorylates MST2 on Ser15, Ser18, and Ser516, and disrupts PP1γ from the complex (Figure 5) [297]. The mechanism by which these phosphorylations abolish PP1γ interaction with NEK2A is currently unknown. The absence of PP1γ leaves NEK2A-mediated phosphorylations of linker proteins unopposed, thereby allowing pre-mitotic resolution of centrosomes and consequently minimizing the rate of erroneous interactions that the mitotic spindle might establish with chromosomes [302,303,304,305].

Phosphatases not only control the assembly of the mitotic spindle, but also its orientation. The position that the spindle assumes in mitosis defines the axis of cell division and plays a key role in cell fate determination in tissues. Depletion of PP4 from progenitor cells of the mouse neocortex causes premature differentiation into neurons and severe defects in cortical layering and brain cytoarchitecture [306]. Biochemical and functional studies demonstrated that PP4-mediated dephosphorylation of NDEL1 at CDK1 sites promotes its binding to LIS1-Dynein to efficiently generate microtubule pulling forces that orient the spindle in parallel to the neuroepithelial surface [306]. This mechanism was shown to be critical at the onset of neurogenesis to ensure extensive symmetric cell division required for the expansion of the progenitor pool [306]. The orientation of the mitotic spindle is also regulated upstream by the distribution of the conserved Gαi-LGN-NuMA module at the cell cortex [307,308,309,310]. Cortical NuMA recruits Dynein complexes that interact with the plus ends of astral microtubules and generate pulling forces to orient the spindle apparatus [311,312,313]. Hence, the localization of NuMA must be tightly controlled, as both insufficient and excessive levels at the cortex impair accurate spindle orientation [312,314]. The association of NuMA with the cell cortex is dynamically modulated by concerted action of CDK1-Cyclin B, Aurora A, PLK1, and several phosphatases. Prior to anaphase onset, phosphorylation of NuMA by Aurora A directs it from the spindle poles to the cortex [315,316], whereas phosphorylation by PLK1 and CDK1-Cyclin B displaces NuMA from the cortex [317,318]. Importantly, phosphorylation of Thr2055 by CDK1 is continuously antagonized by PP2A activity. This results in moderate levels of cortical NuMA during metaphase to ensure accurate spindle positioning and contributes to the enhancement of its cortical recruitment following CDK1 inactivation, which is required to sustain robust spindle elongation in anaphase [317,319]. Protein phosphatase 1-α in complex with its regulatory subunit Repo-Man also promotes cortical accumulation of NuMA in anaphase. This occurs, however, through a separate pathway that is kept in check during metaphase by p37/UBXN2B, a cofactor of the p97 AAA ATPase [320]. Although exactly how p37/UBXN2B limits PP1α-Repo-Man activity towards NuMA is unknown, this balance is critical to prevent excessive Dynein-dependent forces and concomitant spindle orientation defects. Future studies are also expected to cast new light on the mechanism by which PP1α-Repo-Man controls NuMA cortical localization. Moreover, at least in *C. elegans*, PP6 was also shown to positively regulate pulling forces during anaphase by promoting the cortical localization of LGN and NuMA homologues [316,321]. However, the underlying mechanism remains elusive.

### 2.4. Kinetochore-Microtubule Attachments: PP1 and PP2A-B56 Keep Them Stable

To ensure faithful genome partitioning, sister kinetochores must become attached to microtubules emanating from opposite spindle poles. This attachment configuration is termed amphitelic or bioriented and enables the dividing cell to segregate sister chromatids towards opposite sides during anaphase and in this way distribute one copy of each chromosome per daughter nucleus. Amphitelic attachments assume an end-on arrangement with the plus-end of microtubules embedded within the kinetochore. This is believed to provide sufficient stability to sustain the force released by the depolymerisation of kinetochore-associated microtubules during anaphase [322]. However, the initial contacts occurring between kinetochores and the spindle microtubules are asynchronous and stochastic, which implies that amphitelic attachments are often preceded by immature or erroneous microtubule interactions. Following nuclear envelope breakdown, kinetochores might establish lateral (kinetochores interact with the side of microtubules), syntelic (both kinetochores bind microtubules from the same spindle pole), or merotelic attachments (one kinetochore binds microtubules that originate from both spindle poles) [303,323,324,325,326,327,328]. Consequently, accurate chromosome segregation requires the destabilization of the inadequate kinetochore-microtubule interactions and the formation of robust amphitelic end-on attachments, events that are orchestrated by the coordinated activities of molecular motors, kinases, and phosphatases [329,330,331].

A central player in the formation of stable end-on attachments is the NDC80 complex at the outer-kinetochore, which directly binds microtubules through the CH domain and the N-terminal tail of NDC80/HEC1 [332,333,334]. The CH domain encompasses several positively charged residues that contact both α- and β-tubulin at the intra- and inter-tubulin interfaces [333,334,335]. The highly basic nature of the structurally disordered N-terminal tail further increases the affinity for microtubules by directly interacting with negatively charged E-hooks of tubulin subunits [335]. This binding arrangement and the oligomerization of the NDC80 complex confers the kinetochore the capacity to maintain a processive association with dynamic microtubules and in this way couple chromosome movement to microtubule depolymerisation/polymerization [327,335]. The interaction between NDC80 and tubulin is, however, destabilized by Aurora B-mediated phosphorylation of multiple residues on the NDC80 N-terminal tail. The negatively charged phosphates neutralize the intrinsic positive charge of the NDC80 N-terminus, significantly reducing its binding affinity for microtubules [332,333,334,335,336,337,338,339,340]. Notably, this destabilization mechanism acts preferentially on incorrect kinetochore-microtubules attachments. Although the basis for this selectivity is not fully understood, the prevailing model is that the level of tension across centromeres and/or kinetochores plays an important role in controlling the capacity of Aurora B to phosphorylate NDC80 [341,342,343,344]. Aurora B is the catalytic subunit of the chromosome passenger complex (CPC) that also comprises INCENP, Survivin, and Borealin as scaffold subunits. Survivin and Borealin mediate CPC recruitment to centromeres, which subsequently promotes Aurora B T-loop activation through trans-autophosphorylation [345,346,347,348,349,350,351,352]. Microtubules responsible for amphitelic attachments impose tension or alterations on the kinetochore architecture so that the distance between centromeric active Aurora B and the outer kinetochore increases enough to preclude NDC80 phosphorylation (Figure 6A–D) [353,354,355,356]. On the other hand, erroneous attachments fail to produce sufficient tension, thus leaving NDC80 within reach of Aurora B activity to ultimately cause microtubule detachment [342,357,358,359]. Because unattached kinetochores are tensionless, this error correction mechanism might seemingly represent a problem for the formation of initial end-on microtubule attachments (Figure 6A). In that respect, it is important to consider that the early contacts that kinetochores establish with microtubules are predominantly lateral and that PP2A-B56 at outer-kinetochores was shown to play a critical role in antagonizing Aurora B dependent phosphorylations (Figure 6B). The interaction of kinetochores with the side of microtubules is mediated by the molecular motors, Dynein and CENP-E, and are therefore impervious to the phosphorylation state of NDC80 [303,324,325,327,328,360]. Dynein minus-end motor activity transports chromosomes towards the spindle pole, exposing the kinetochores to a higher density of microtubules that might favor direct end-on capture [361,362,363,364]. As spindle microtubules become more stable and detyrosinated, CENP-E motility increases to promote the movement of the persisting laterally attached chromosomes towards the equator [325,365,366,367,368,369]. Once it reaches microtubule ends, CENP-E converts from a lateral transporter into a microtubule tip-tracker that tethers kinetochores to the dynamic microtubule tips, thus directly contributing for the formation and stability of end-on attachments [370,371,372]. Importantly, lateral attachments exert intermediate levels of tension that increase the distance between Aurora B and the outer-kinetochore. This concurs with the dephosphorylation of the NDC80 N-terminal tail by PP2A-B56γ to allow the formation of stable end-on attachments [360,373,374,375] (Figure 6B–D). The recruitment of PP2A-B56γ to the outer-kinetochore is directly mediated by a LxxIxE motif within the KARD domain of BUBR1 [66,69,373,374,375]. A conserved pocket in the B56γ regulatory subunit recognizes and binds to the LxxIxE motif [65,66,67,68] (Figure 2C). Phosphorylations within and around the LxxIxE motif, respectively catalyzed by CDK1 and PLK1, significantly strengthen the interaction [66,67,373,374,376,377]. Notably, although the binding pocket is conserved in all B56 isoforms, BubR1 binds preferentially to B56γ and possibly to B56δ and presents limited association with B56α and B56ε, which instead display preference for centromeric SGO2 [69,378,379,380]. Similarly, several other mitotic proteins harboring LxxIxE motifs were shown to bind preferentially to B56γ over B56α [69]. A possible explanation for B56 differential binding, and therefore localization, was recently attributed to specific differences in a small loop at the C-terminus of each B56 isoform. The identity of four critical amino acids within the loop seems to modulate its capacity to repress or allow binding to LxxIxE motifs [69]. The residues, Glu405, Pro409, Val412, and Ala413, define the EPVA loop of B56α and B56ε subunits, which not only hampers binding to LxxIxE motifs, but is also critical to enable B56α and B56ε interaction with SGO2. Remarkably, replacing the EPVA signature amino acids by the corresponding residues of the B56γ isoform (Thr405, Lys409, His412, and Gly413) enhanced B56α ability to bind BUBR1 and other LxxIxE-containing proteins [69]. Although structural details on how the C-terminal loop controls B56 specificity remain unknown, these observations clearly demonstrate sub-functionalization between the B56 isoforms. Whether such specialization allows PP2A activity to be differently regulated in specific subcellular compartments represents an alluring possibility that could be addressed in future studies.

Because BUBR1 associates with KNL1 in the absence of microtubules, the binding of PP2A-B56γ to BUBR1 places the phosphatase in the close proximity of NDC80 to efficiently counteract Aurora B destabilizing activity and enable initial end-on microtubule interactions [373,374,375,381] (Figure 6A,B). As microtubule occupancy increases, the outer-kinetochore is further pulled away from Aurora B zone of influence and attachments stability concomitantly increases. At this point, PP1 activity at kinetochores increases to silence the mitotic checkpoint (as discussed in the following section) and ensure that mature end-on microtubule attachments remain stabilized [360,382,383,384] (Figure 6C,D). Recruitment of PP1 to kinetochores is mediated in part by SILK and RVSF motifs on KNL1 N-terminus [382]. At tensionless kinetochores, both PP1-docking motifs are phosphorylated by Aurora B, which significantly represses their ability to interact with PP1 [382] (Figure 6A). As end-on attachments mature, KNL1 N-terminus is no longer accessible to Aurora B and PP2A-B56γ dephosphorylates SILK and RVSF to allow PP1 binding (Figure 6B–D).

The PP1γ-KNL1 complex further suppresses kinetochore Aurora B activity, hence enhancing the stability of amphitelic attachments [385,386]. Disrupting the PP1γ-KNL1 interaction destabilizes kinetochore-microtubule attachments on metaphase-aligned chromosomes while failing to cause a discernible impact on chromosome congression [360,382]. Thus, although PP1γ-KNL1 activity seems to be dispensable for initial microtubule binding, it does contribute to preserve attachment stability following biorientation. In that respect, PP1γ-KNL1 antagonizes the activating T-loop autophosphorylation of Aurora B [385,387,388] and enables the recruitment of the SKA complex to the KMN network [389,390] (Figure 6C,D). The SKA complex at the kinetochore-microtubule interface is proposed to enhance the microtubule-binding affinity and end-tracking ability of the Ndc80 complex [391]. This increases the stability of attached microtubule bundles and maintains the kinetochore associated with dynamic microtubules, thus enhancing the ability of end-on attachments to sustain load-bearing force and couple chromosome movement to microtubules plus-end dynamics [389,391,392,393,394,395,396,397,398,399]. Aurora B-mediated phosphorylation of the SKA complex reduces its binding affinity for NDC80, which might provide a safety strategy to minimize premature stabilization of erroneous attachments during early prometaphase [389,391,396,400,401]. As chromosomes congress and PP1γ-KNL1 activity increases, SKA recruitment is triggered and the complex becomes maximally enriched at amphitelically attached kinetochores [394,396,400,402]. In *C. elegans*, timely association of the SKA complex with bioriented kinetochores is facilitated by dephosphorylation of the NDC80 N-terminal tail [389]. Whether PP1γ-KNL1 cooperates with PP2A-B56γ to dephosphorylate Aurora B phosphosites on the NDC80 tail remains unclear, as well as PP1γ-KNL1 aptitude to directly target Aurora B-mediated phosphorylations of the SKA complex. Interestingly, the SKA complex directly binds PP1γ through the C-terminal domain (CTD) of its SKA1 subunit and independently of KNL1, thus representing an additional pathway to increment PP1γ at kinetochores of bioriented chromosomes [403]. Although PP1γ-SKA significantly contributes to timely inactivation of the mitotic checkpoint, its role in kinetochore-microtubule attachments is unclear [403]. The accumulation of PP1 on bioriented kinetochores can be further endorsed by spindle motors. The budding yeast Kinesin 5, CIN8, associates with PP1/GL7 through a canonical RVxF motif [404]. Kinetochore localization of CIN8 dramatically increases during metaphase in a microtubule-dependent manner and is mediated by a direct interaction with the DAM1 complex, a functional analog of the SKA complex in vertebrates [393,395,404]. Binding to DAM1 places the PP1/GL7-CIN8 complex near the N-terminus of NDC80, likely resulting in dephosphorylation of its N-terminal tail. This increases the affinity of NDC80 for microtubules, and consequently, the stability and tension required for robust attachments [404]. Human cells engage in a similar strategy, but differ in the choice of motor, as the human Kinesin-5 orthologue does not localize at kinetochores. Instead, PP1γ binds to RVxF motifs on the plus-end directed motors, KIF18A [405,406] and CENP-E [407]. Pools of PP1γ-KIF18A and PP1γ-CENP-E accumulate at the ends of kinetochore-attached microtubules as cells progress from prometaphase to metaphase, thus providing a localized delivery of PP1γ to oppose NDC80 phosphorylation and promote attachment stability [407,408]. Importantly, Aurora A-mediated phosphorylation of CENP-E on its PP1-docking motif averts premature binding of PP1γ. This phosphorylation is maximal on polar chromosomes and is required for CENP-E-powered congression along kinetochore-microtubule bundles [407]. As chromosomes move away from Aurora A, dephosphorylation of CENP-E takes place, enabling PP1γ recruitment. Although it is presently uncertain which phosphatase pool dephosphorylates the PP1-docking motif on CENP-E, this was shown to be essential for the formation of stable end-on attachments and biorientation of chromosomes that congressed from the spindle poles [407].

Several other PIPs have been shown to act as important kinetochore-targeting factors of PP1, but their precise impact on microtubule attachment has been difficult to discern. For instance, PP1 bound to MYPT1 was shown to antagonize the phosphorylated T-loop of PLK1, which consequently renders the kinase inactive [409,410]. Contrasting with the previously described PP1-PIP complexes, MYPT1-PP1 associates with kinetochores preferentially during early mitosis and depends on CDK1-Cyclin A activity [45]. CDK1-Cyclin A-mediated phosphorylation of MYPT1 promotes its interaction with PLK1, hence dampening PLK1 activity and therefore the stability of kinetochore-microtubule attachments [45]. This is thought to generate a temporally-defined microtubule-destabilizing environment to facilitate the correction of erroneous attachments typically produced during early prometaphase [411]. The observation that PLK1 activity is suppressed prior to chromosome biorientation seems somehow difficult to reconcile with its requirement for initial end-on attachments. However, it is becoming increasingly clear that PLK1 modulates the stability of kinetochore-microtubule interactions through multiple kinetochore/centromere pools and possibly responding to different inputs [373,376,412,413,414,415,416,417,418,419,420,421,422,423,424]. The conserved regulatory subunit, SDS22/PPP1R7, has been reported to be essential for the activity of multiple PP1 isoforms at kinetochores. Depletion of SDS22/PPP1R7 in human cells perturbs kinetochore-microtubule attachments and results in chromosome alignment defects [385,386]. Paradoxically, SDS22/PPP1R7 does not seem to localize at kinetochores and PP1 fails to dephosphorylate its kinetochore substrates while bound to SDS22/PPP1R7 [425]. Critical insight into this apparent discrepancy came from the observation that SDS22 and I3 transiently act as chaperones during PP1 biogenesis [426]. Binding of SDS22 and I3 stabilizes newly-synthetized PP1 in an inactive state until it is under control of substrate specifier PIPs [425,426,427,428]. The p97 AAA-ATPase complex subsequently catalyzes the disassembly of the heterotrimeric complex, which enables PP1 to associate with one of its numerous adaptor subunits and form a functional holoenzyme [426].

### 2.5. Silencing the Spindle Assembly Checkpoint: PP1 and PP2A-B56 Shut It Up

To ensure that each daughter nucleus receives one copy of the genome, chromosome segregation can only occur after all sister-kinetochore have established stable amphitelic attachments. Timely control of anaphase onset is achieved through the spindle assembly checkpoint (SAC), a conserved signaling pathway primarily instated by unattached/tensionless kinetochores that inhibits the ubiquitin-dependent proteolysis of Securin and Cyclin B. Degradation of Securin triggers the cleavage of centromeric cohesion by the protease Separase to enable sister chromatid separation, whereas reduced levels of Cyclin B lead to inactivation of CDK1, which consequently reverts the cell into an interphase state. According to the prevailing mechanistic model of SAC signaling, unattached/tensionless kinetochores catalyze the assembly of a diffusible protein tetramer known as mitotic checkpoint complex (MCC) that binds to the E3 ubiquitin ligase anaphase promoting complex/cyclosome (APC/C) and thereby prevents it from targeting Securin and Cyclin B for degradation [429,430,431,432]. A weakened SAC signaling was shown to allow anaphase onset in the presence of one or few unattached chromosomes, which resulted in varied levels of aneuploidy [433,434,435,436]. On the other hand, failing to switch off the SAC in a timely manner delays the cells in metaphase, thus increasing the likelihood of merotelic attachments and cohesion fatigue, errors that often lead to lagging chromosomes and chromatin bridges [436,437]. Hence, to safeguard against genome imbalances, the SAC must be robust enough so that a single unattached kinetochore is sufficient to halt the transition to anaphase, but also highly responsive to allow swift mitotic exit upon formation of stable microtubule attachments [384,438,439]. Activation and strength of SAC signaling is driven by the activity of multiple mitotic kinases, with monopolar spindle 1 (MPS1) assuming the leading role [440,441]. Unsurprisingly, counteracting phosphatases play a critical role in extinguishing the checkpoint signal and ensure timely cell cycle progression.

Monopolar spindle 1 associates with unattached kinetochores and phosphorylates KNL1 on its MELT repeats, creating docking sites for the hierarchical recruitment of additional SAC proteins necessary for MCC assembly [442,443,444,445,446] (Figure 6A). Heterodimers of BUB1-BUB3 bind to phosphorylated MELTs and recruit BUBR1-BUB3 through hetero-dimerization [447]. Monopolar spindle 1 phosphorylates KNL1-associated Bub1 to promote the binding of MAD1-MAD2 complexes [448,449,450,451,452,453]. Subsequently, MPS1-mediated phosphorylation of MAD1 dramatically accelerates the structural conversion of MAD2 into a conformer that binds to CDC20, the rate-limiting step for MCC assembly [450,451]. These multi-target phosphorylations along the SAC signaling cascade render the checkpoint highly responsive to variations in MPS1 activity [454]. Activation of MPS1 is triggered during early mitosis by T-loop trans-autophosphorylation. The CH domains of NDC80 and NUF2 directly mediate the recruitment of MPS1 to unattached kinetochores [455,456], which likely favors the kinase trans-autoactivation [457,458,459,460,461,462]. Kinetochore tethering of MPS1 and its activity are enhanced by Aurora B, PLK1, and CDK1, but the underlying mechanisms remain largely unknown [463,464,465,466,467,468,469,470]. On the other hand, MPS1 promotes centromeric accumulation of Aurora B via the BUB1-H2A-SGO1 axis and was shown to directly foster Aurora B activity, thus setting a positive feedback loop that ensures prompt and robust activation of both kinases at the onset of mitosis [471,472].

Once all kinetochores attach to microtubules and become stably bioriented, the SAC must be silenced to permit APC/C activation and initiate anaphase. This primarily entails mechanisms that prevent and revert the phosphorylations catalyzed by MPS1. The region of the NDC80 CH domain that interacts with MPS1 partially overlaps with the microtubule binding interface [455], which significantly restrains MPS1 ability to associate with kinetochores at a high microtubule occupancy. This biochemical competition imposes a severe reduction on the amount of MPS1 present at kinetochores once stable end-on attachments are established [455,456] (Figure 6A–D). This concurs with increased kinetochore localization of PP1, which in *Drosophila* cells was shown to dephosphorylate the T-loop of MPS1 to ensure the inactivation of residual MPS1 molecules remaining at kinetochores and thereby efficiently avert the phosphorylation of SAC substrates during metaphase [473] (Figure 6C,D). PP2A-B55 was also shown to dephosphorylate MPS1 in MEFs [174], but the functional relevance of the targeted residue remains unclear [460,465].

Suppressing MPS1 activity at bioriented kinetochores is not sufficient to promptly silence the SAC, as the phosphorylations carried out to instate the checkpoint still need to be reversed. This task has been almost consensually attributed to PP1 [379,442,474,475,476,477,478,479,480]. Several PP1-PIP complexes were shown to contribute for SAC silencing. PP1γ bound to KNL1 is particularly well positioned, as it lies adjacent to the site of MCC production. Impairing PP1γ interaction with KNL1 prevents the dephosphorylation of MELT motifs and delays exit from mitosis in different organisms [379,442,478,479,480,481]. Human cells lacking the PP1γ-KNL1 complex and incubated with microtubule depolymerizing drugs were able to remain in mitosis even after MPS1 inhibition, suggesting that PP1γ directly targets the phospho-MELTs to cease the recruitment of SAC proteins and consequently silence the SAC [379]. Efficient binding of PP1γ to KNL1 relies on the dephosphorylation of SILK and RVxF motifs by PP2A-B56γ in complex with BUBR1, which is itself indirectly recruited to KNL1 upon MELT phosphorylation [379,382,442,443,444] (Figure 6A–C). This negative feedback loop enables the checkpoint to prime its own inhibition, thus rendering signaling quenching highly responsive to microtubule binding [380]. Prompt SAC silencing following end-on attachment is further ensured by pools of PP1γ delivered to kinetochores through microtubules [403,478,482] (Figure 6C,D). In fission yeast, DIS2/PP1 bound to the kinesin-8 motor KLP5-KLP6 heterodimer cooperates with SPC7/KNL1-associated DIS2/PP1 in switching off the SAC [478]. However, the kinetics of checkpoint silencing mediated by each pathway differs considerably, with DIS2-SPC7 acting firstly under low microtubule occupancy, while DIS2-KLP5-KLP6 appears to operate during later time points as attachments stabilize [478,482]. Preventing PP1γ binding to the KIF18A orthologue in human cells delays the transition to anaphase, but whether this is caused by inefficient SAC silencing is unclear as the activity of PP1γ-KIF18A is required for the stability of kinetochore-microtubule attachments [406,408]. Likewise, disruption of the interaction between PP1γ and SKA1 through deletion of SKA1 CTD results in a prolonged metaphase delay [403]. Notably, replacing the CTD by a PP1-binding motif or by a direct fusion to PP1γ largely restores timely anaphase onset, suggesting that the recruitment of PP1γ mediated by the SKA complex is important to efficiently silence the SAC at bioriented kinetochores [403]. However, SKA1 CTD has been shown to be required for the stability of kinetochore-microtubule attachments [391,401], which calls for some caution in ascribing a direct role for PP1γ-SKA1 in switching off the SAC. It is possible that PP1γ-SKA1 might instead counteract destabilizing phosphorylations on NDC80 and thereby indirectly promote SAC silencing by increasing the stability of end-on attachments [384]. In line with that, the SKA complex was shown to directly interact with multiple regions of NDC80 [389,390,399,483,484,485]. On the other hand, this also means that PP1γ is favorably positioned to dephosphorylate and inactivate MPS1 when robust microtubule attachments are established [384]. It is therefore important to determine which specific phosphorylations are targeted by each pool of PP1γ present at kinetochores. This will be critical to better understand their role and how the activities of PP1γ-KNL1 and PP1γ complexes recruited through microtubules are coordinated to ensure efficient SAC silencing. In that respect, recent structural data have shown that PP1-docking motifs in KNL1 overlap extensively with domains that also mediate binding to microtubules, and, at least in vitro, these are mutually exclusive [486]. This hints that in mammalian cells, as described for fission yeast [478,482], PP1γ-KNL1 operates mainly at a low microtubule occupancy earlier in mitosis and sets up the recruitment of microtubule-dependent pools of PP1γ to bioriented kinetochores. This would be consistent with the observation that KNL1 mutants defective in PP1γ binding fail to switch off the SAC in nocodazole [379], while exhibiting only a modest metaphase delay under unperturbed conditions [464], whereas loss of the PP1γ-SKA1 complex results in a prominent SAC-dependent metaphase arrest, but only marginally affects mitotic exit when MPS1 is inhibited in the absence of microtubules [384,403].

Interestingly, PP2A-B56γ bound to BUBR1 might also directly contribute to dephosphorylate phospho-MELT motifs [487] and was recently shown to antagonize MPS1-mediated phosphorylation of BUB1 Thr461 [452]. This phosphorylation stimulates MAD1 binding [451,452,453] and appears to be only transiently required for SAC initiation immediately upon nuclear envelope breakdown [452]. Dephosphorylation of BUB1 Thr461 is triggered during early prometaphase concomitantly with the recruitment of PP2A-B56γ to BUBR1. Phosphorylation of BUB1 Thr461 is virtually undetectable in metaphase or after a drug-elicited prometaphase arrest, suggesting that BUB1 dephosphorylation by PP2A-B56γ is impervious to kinetochore attachment status [452]. Thus, MPS1 and PP2A-B56γ-BUBR1 establish an incoherent feedforward loop that restricts BUB1 Thr461 phosphorylation to the period of time required to assemble PP2A-B56γ-BUBR1 at kinetochores [452]. Because this transient pulse in BUB1 Thr461 phosphorylation is not affected by microtubule occupancy, it was proposed to function as a biochemical timer defining the minimum length of time the cell spends in mitosis [452]. Bub1 further promotes APC/C inhibition through an MCC-independent mechanism. Kinetochore BUB1 acts as a scaffold for PLK1-mediated phosphorylation of the APC/C co-activator, CDC20. Although the mechanistic underpinnings remain elusive, this phosphorylation was shown to inhibit CDC20 ability to bind and activate the APC/C [488]. The inhibitory phosphorylation was proposed to be removed by PP2A-B56γ-BUBR1 [488]. However, CDC20 phosphorylation remained more stable in cells incubated with okadaic acid than in cells depleted of all B56 isoforms, suggesting that PP1 or additional phosphatases might contribute to dephosphorylate CDC20 and enable APC/C activation [489,490]. Further inhibition of CDC20 is promoted by CDK1-Cyclin B, which paradoxically also phosphorylates the APC/C to ensure its proper ubiquitination activity [64,489,490,491,492]. PP2A-B55 has been implicated in opposing the inhibiting and activating CDK1 phosphorylations on CDC20 and APC/C, respectively [64]. The intrinsic preference of PP2A-B55 towards pThr over pSer is proposed to provide a short period of time in which dephosphorylated CDC20 is able to interact with phosphorylated and functional APC/C [64]. However, the APC/C-CDC20 is known to be active prior to reactivation of PP2A-B55 [493], which in fact requires APC/C-CDC20 dependent repression of GWL activity. Hence, removal of CDK1 inhibitory phosphorylations on CDC20 likely relies on the activity of other phosphatases, an undertaking recently attributed to PP1γ-KNL1 in *C. elegans* [491].

### 2.6. Mitotic Exit: PP1 and PP2A-B55 Clear the Way

Once the SAC is switched off and APC/C-CDC20-dependent degradation of Securin and Cyclin B occurs, the cell becomes irreversibly committed to exit mitosis. Resumption of an interphase state requires remodeling of the spindle architecture, reassembly of the functional nuclear envelope, and decondensation of chromatin [86]. This extensive structural reorganization relies not only on the inactivation of CDK1, but also on the removal of mitotic phosphorylations, an assignment that in animal cells is primarily led by PP2A-B55 and PP1 phosphatases [58,86]. A key event during anaphase is the reorganization of spindle microtubules into a structure referred to as the central spindle. This is assembled between the segregating chromosomes to define the position of the cleavage furrow, thus ensuring spatial and temporal coordination of cytokinesis with genome partitioning [86,494,495,496,497]. Formation of the central spindle requires PRC1, a microtubule-bundling protein that stabilizes and organizes microtubules in an antiparallel overlapped fashion and promotes anaphase spindle elongation [498]. Phosphorylation of PRC1 by CDK1-Cyclin B prevents its microtubules’ bundling activity before anaphase [499,500,501,502]. Timely PRC1 dephosphorylation is catalyzed by PP2A-B55 [164] to enable PRC1 interaction with the kinesin motor KIF4 and its consequent translocation to the plus ends of antiparallel interdigitating microtubules [498,503] (Figure 7A). Furthermore, PP2A-B55-mediated dephosphorylation promotes PRC1 homodimerization that is required for its microtubule bundling activity and thereby for the assembly of the central spindle [164,502,504,505] (Figure 7A).

Reformation of the nuclear envelope and associated nuclear pore complexes around chromatin occurs at the end of mitosis to establish functional interphase nuclei [506,507,508]. Disassembly of the nuclear envelope and of nuclear pore complexes at mitotic entry is triggered by a myriad of phosphorylations catalyzed by several mitotic kinases (Figure 7B). Phosphorylation of Lamins by CDK1 promotes depolarization of the nuclear lamina, the filamentous protein meshwork underlying the nuclear membranes [509,510]. Cyclin-dependent kinase 1 further cooperates with PLK1 in the phosphorylation of several nucleoporins to elicit disintegration of nuclear pore complexes [511,512,513,514,515,516,517,518]. Moreover, phosphorylation of the chromatin binding protein, BAF, by VRK reduces its affinity for chromatin and towards the LEM family of inner nuclear membrane proteins interacting with the nuclear lamina [519,520,521]. Loss of the BAF-mediated link between chromatin and the inner nuclear membrane at mitotic entry is proposed to contribute to nuclear envelope disassembly. Phosphorylations on BAF are however removed during mitotic exit by action of PP2A-B55 (Figure 7B). This restores BAF interaction with chromatin and with components of the nuclear periphery, which was shown to be required for timely nuclear envelope reformation in worms, flies, and human cells [522,523]. Furthermore, recent findings in *Drosophila* cells suggest that PP2A-B55 might also target Lamin and NUP107 to direct their assembly into the nuclear envelope during mitotic exit [523]. Further evidence supporting PP2A-B55 involvement in nuclear pore reformation come from the observation that the phosphatase dephosphorylates NUP153 (as well as NUP107) to promote its recruitment to chromatin in human cells [58] (Figure 7B).

Several findings indicate that PP1 is equally important for reassembly of functional nuclear envelopes. Binding to AKAP149 localizes PP1 on the assembling nuclear envelope, where it dephosphorylates Lamin B to promote its polymerization and accelerates nuclear lamina assembly [524,525,526] (Figure 7B). The regulatory subunit, Repo-Man, targets PP1γ to the periphery of late anaphase chromosomes, where the initial reassembly events take place [508,527,528,529]. There, the N-terminus of Repo-Man directly mediates the recruitment of Importin-β to control early nuclear envelope formation and nucleoporin deposition [528,530,531,532,533,534] (Figure 7B). Binding of Importin-β to Repo-Man does not require PP1γ catalytic activity and is repressed during mitosis by CDK1 phosphorylations on the Repo-Man N-terminus [528]. Localized activity of PP1 seems, however, to be required for later steps of nuclear pore complex formation. One subset of the nucleoporins that is controlled by Importin-β is the NUP107–NUP160 complex [535], which is recruited to chromatin by the nuclear pore complex assembly factor, ELYS/MEL-28 [536,537,538,539,540]. This requires RanGTP-dependent relief of NUP107–NUP160 nucleoporins from an inhibitory association with importin-β, most likely docked at the vicinity of ELYS/MEL-28 by Repo-Man [527,528,529]. This results in the formation of chromatin-bound pre-pores onto which additional nucleoporins are subsequently assembled, a process that in *C. elegans* was recently shown to require the contribution of PP1c/GSP-2 bound to ELYS/MEL-28 [541]. However, direct evidence for dephosphorylation events catalyzed by PP1c/GSP-2 and a mechanistic understanding of its role in the reassembly of nuclear pore complexes are still missing.

As the nuclear envelope reassembles, the segregated chromatin decondenses to re-establish an interphase structure compatible with DNA replication and transcription [542]. This relies on the activity of PP1γ, which, prior to its accumulation at the periphery of late anaphase chromosomes, is homogenously distributed over chromatin during early stages of anaphase. This initial recruitment of PP1γ to chromatin is also mediated by the regulatory subunit, Repo-Man, although through a separate C-terminal module that is dispensable for the peripheral localization [528,543,544]. Analogously to inhibitory phosphorylations on the Repo-Man N-terminal domain, phosphorylation of the C-terminus chromatin-binding module by CDK1 and Aurora B prevents Repo-Man binding to histones before anaphase onset [528,545]. PP1γ-Repo-Man antagonizes the mitotic phosphorylation of Thr3 and Ser10 on histone H3, respectively catalyzed by Haspin and Aurora B [528,546,547]. Dephosphorylation of Thr3 allows the translocation of the CPC from chromosomes to the central spindle, whereas dephosphorylation of Ser10 triggers decondensation and enables the re-association of HP1 to H3K9me3 after mitosis [546,548]. Intriguingly, depletion of Repo-Man fails to cause a dramatic impairment of chromatin decondensation [528]. Because the recruitment of PP1γ to anaphase chromatin is also mediated by Ki-67 [549,550], it is possible that the pool of PP1γ-Ki-67 compensates for the loss of Repo-Man [542]. However, the exact contribution of PP1γ-Ki-67 holoenzyme for chromatin decondensation remains to be demonstrated. Another chromatin targeting factor for PP1γ is the regulatory subunit, PNUTS. Binding of PP1γ to PNUTS through a canonical RVxF motif was shown to enhance chromatin decondensation in vitro [551]. However, the association of PP1γ-PNUTS with chromatin is only detected during telophase, after H3 Ser10 dephosphorylation and nuclear envelope reformation [551]. Therefore, the precise targets of PP1γ-PNUTS and its relevance for chromatin decondensation are still elusive.

## 3. Spatiotemporal Control of Phosphatase Activities

The accuracy of mitosis requires tight spatiotemporal control of phosphatase activities. Although PP1 does contribute to mitotic exit progression, the bulk of CDK1-mediated phosphorylations are removed by PP2A when in complex with B55 [58]. Thus, PP2A-B55 must be kept inhibited from nuclear envelope breakdown to anaphase onset for the sake of mitotic compliance, becoming then reactivated to orchestrate the major events underlying anaphase and mitotic exit. As mentioned in Section 2.1, repression of PP2A-B55 is triggered by CDK1-mediated activation of GWL, which subsequently phosphorylates ENSA and Arpp19 to promote their inhibitory binding to PP2A-B55 [165,166,176,177] (Figure 3A). Both inhibitors are high affinity substrates of PP2A-B55, but because their dephosphorylation rate is orders of magnitude lower than CDK1-phosphorylated targets, this efficiently restrains PP2A-B55 activity in what is known as “unfair competition” [552]. Degradation of Cyclin B upon transition into anaphase initiates the cascade that activates PP2A-B55. Decreasing CDK1-Cyclin B activity is proposed to render PP1 active to dephosphorylate and partially inactivate GWL [553,554,555,556] (Figure 3A). This means that the phosphorylations that are ultimately removed from ENSA and Arpp19 by PP2A-B55 now fail to be efficiently replaced, hence allowing PP2A-B55 to target CDK1-phosphorylated substrates [552]. In that respect, PP2A-B55 further dephosphorylates GWL to ensure its complete inactivation and thereby maintain elevated PP2A-B55 activity to promote mitotic exit [553,554,555,556] (Figure 3A).

Importantly, active PP2A-B55 seems to follow a timely defined dephosphorylation program so that events in anaphase, telophase, and cytokinesis are carefully coordinated and regulated to ensure a successful cell division [58,164]. It is absolutely critical that dephosphorylation of PRC1 and of other spindle proteins occurs before the dephosphorylation of Lamins and of nucleoporins, so that the nuclear envelope and nuclear pores can only assemble and import proteins once chromosomes are successfully segregated [58,59]. The timing of substrate dephosphorylation during mitotic exit is determined by the net basic charge of the two polybasic regions upstream and downstream of the CDK1-phosphorylated site. Spindle proteins have pronounced polybasic regions and are therefore readily selected and dephosphorylated by PP2A-B55 early in anaphase, whereas nuclear envelope proteins comprise an inferior number of basic residues and are therefore dephosphorylated at lower rates. This differential catalytic efficiency enables PP2A-B55 to achieve temporal dephosphorylation of CDK1 sites during mitotic exit and, thereby, coordinate the segregation of chromosomes with nuclear envelope reformation [58]. Moreover, the intrinsic preference of the PP2A catalytic subunit towards pThr over pSer residues provides an additional regulatory element to ensure ordered dephosphorylation of substrates during mitotic exit [63,64]. Early anaphase events seem to be predominantly driven by pThr dephosphorylation, whereas pSer-bearing substrates are dephosphorylated with slower dynamics, thus contributing for sequential execution of the mitotic exit program [64].

The mechanism controlling PP1-mediated inactivation of GWL remains, however, somewhat contentious. It has been proposed that PP1 activity is globally repressed in mitosis by a CDK1 inhibitory phosphorylation on its C-terminus [557,558,559] and by binding to I-1, an inhibitory PIP whose interaction with PP1 is promoted by a PKA-dependent phosphorylation residue [560]. Notably, both phosphorylations are targeted by PP1 itself. Thus, declining CDK1-Cyclin B activity following SAC silencing is thought to be sufficient to allow PP1 autodephosphorylation and I-1 dephosphorylation, thereby resulting in increased activation of PP1 to levels that drive mitotic exit [553,554,555,556,560]. This sequential activation of PP1 and PP2A-B55 occurring in animal cells resembles a phosphatase relay reported in fission yeast, which also operates as a timer for the orderly dephosphorylation of mitotic substrates at the end of division [559]. Interestingly, not all PP1 molecules are inhibited by CDK1 during mitosis [561] and, therefore, are likely to be involved in the phosphatase relay. In fact, the activity of multiple PP1 holoenzymes is known to be required for the stability of kinetochore-microtubule attachments and efficient SAC silencing, events that occur before APC/C-CDC20-triggered degradation of Cyclin B. It would be interesting to identify which specific pools of PP1 are under control of CDK1-Cyclin B to elucidate how these engage with GWL to instate the phosphatase relay and to understand the mechanism that selectively renders some PP1 holoenzymes impervious to this inhibition. The activity of PP1-PIP complexes operating in mitosis is, however, locally regulated, frequently by controlling PP1 interaction with regulatory subunits or substrates. This generally entails the phosphorylation of RVxF PP1-binding motifs by Aurora B, which abrogates the PIP ability to bind PP1 [562]. This regulatory strategy enables the cell to tightly control the levels of PP1 at centromeres and at kinetochores in response to microtubule attachment status [562,563] (Figure 6A–D). The inhibitory phosphorylations on the KNL1 PP1-docking motif are removed by BUBR1-PP2A-B56, which allows PP1γ to associate with KNL1. This comprises another phosphatase relay, now at kinetochores and required for efficient SAC silencing and timely anaphase onset.

The activity of PP2A-B56 at kinetochores is modulated to avert deleterious dephosphorylation during early mitosis, when correction of erroneous attachments is often required. This is accomplished by BOD1, a small protein that shares sequence similarity with ENSA and Arpp19, and that specifically binds to PP2A-B56 when phosphorylated by CDK1 [564]. BOD1-mediated inhibition of PP2A-B56 is maximal at early prometaphase, possibly to prevent premature dephosphorylation of the NDC80 N-terminus and of KNL1 PP1-docking motifs [565]. As the cell progresses through mitosis, relief of PP2A-B56 from the inhibitory interaction with BOD1 is essential for the formation and stabilization of end-on attachments. The mechanism responsible for BOD1 dephosphorylation remains, however, undetermined.

## 4. Concluding Remarks and Future Directions

In the past decades, we have witnessed the emergence of protein phosphatases as critical regulators of mitosis. In recent years, the list of identified substrates and regulators has increased considerably, as has our understanding of phosphatases’ roles in mitotic progression.

Structural and biochemical determinants of phosphatase holoenzyme assembly and specificity have been unveiled. Research has made important advances in the functional characterization of protein phosphatases and has generated critical molecular knowledge to better understand their spatiotemporal control. Importantly, these achievements have provided new and renovated perspectives on the regulation of mitotic events, whose accurate execution is absolutely critical for genomic stability.

Nonetheless, enduring unanswered questions deserve further investigation. It remains relevant to dissect the regulatory network underlying the activity of CDC25 isoforms. The field will also benefit from a comprehensive identification of the precise substrates that are targeted by specific Ser/Thr phosphatase holoenzymes and of the upstream signaling inputs controlling the subcellular localization and activity of the different holoenzymes. These have been hampered by the inability to routinely tackle specific interactions of phosphatases with substrates or regulators. Further structural insights into SLiM-mediated interactions might reveal critical mechanisms to enable the disruption of specific phosphatase complexes in a way that is compatible with systematic studies. Furthermore, the development of specific inhibitors targeting phosphatase-SLiM contacts should also fuel their potential use as therapeutic strategies in disease. This might represent a valuable alternative, or complement, to protein kinase inhibitors that are often associated with on-target resistance.

## Figures and Tables

**Figure 1 biomolecules-09-00055-f001:**
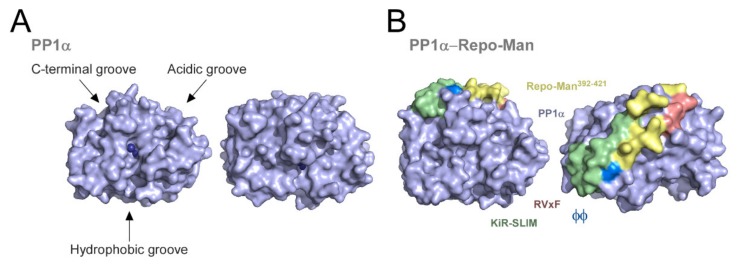
Structure of PP1α catalytic subunit and its interaction with the regulatory subunit, RepoMan. (**A**) Surface representation of the three-dimensional (3D) structure of PP1α catalytic subunit. Manganese atoms in the active site are represented as dark blue spheres and lie at the Y-shaped intersection of the three substrate-binding grooves. Protein Data Bank (PDB) accession identification (ID) for the structure: 4MOV. (**B**) Surface representation of the three-dimensional structure of the PP1α catalytic subunit in complex with Repo-Man^392–421^. The PP1α catalytic subunit is represented in light blue and the Repo-Man^392–421^ peptide is represented in yellow. The interaction is mediated by RVxF (red), ΦΦ (blue) and KiR-SLiM (green) motifs on Repo-Man^392–421^ that bind to specific pockets on the PP1α surface. PDB accession ID 5IOH.

**Figure 2 biomolecules-09-00055-f002:**
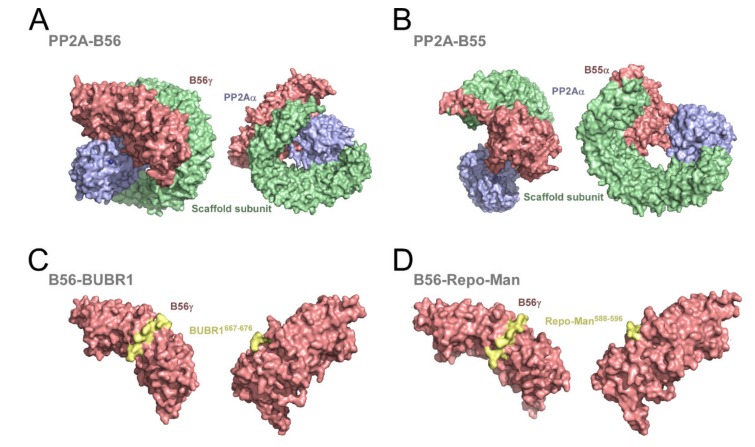
Structure of PP2Aα holoenzymes. (**A**,**B**) Surface representation of the three-dimensional structure of the PP2Aα catalytic subunit (light blue) in complex with the scaffold subunit and the regulatory subunits, B56γ (**A**) or B55α (**B**). The PP2A catalytic subunit is represented in light blue with manganese atoms in the active site represented as dark blue spheres. The scaffold subunit is represented in green and the regulatory subunits are depicted in light red. Protein Data Bank (PDB) accession ID 2NPP (**A**) and 3DW8 (**B**). (**C**,**D**) Surface representation of the three-dimensional structure of B56γ (light red) interacting with BUBR1^667–676^ and Repo-Man^588–596^ peptides (yellow). Both peptides encompass a LxxIxE motif that binds to a conserved basic pocket at the concave surface of B56γ. PDB accession ID 5K6S (**C**) and 5SW9 (**D**).

**Figure 3 biomolecules-09-00055-f003:**
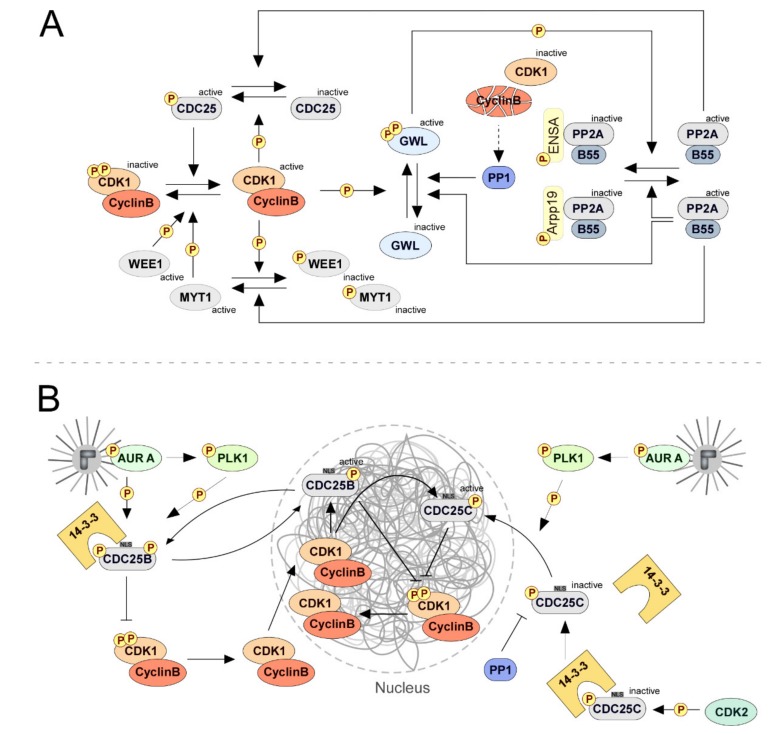
**Molecular circuitry controlling mitotic entry.** (**A**) CDC25 phosphatases dephosphorylate and activate CDK1-Cyclin B to promote mitotic entry. CDC25 activity antagonizes the inhibitory phosphorylations on CDK1 (Thr14 and Tyr15) catalyzed by MYT1 and WEE1. Active CDK1-Cyclin B respectively inhibits and promotes MYT1/WEE1 and CDC25 activities, establishing both a double negative feedback loop and a positive feedback loop. This regulatory circuitry defines a bistable trigger that ensures irreversible transition into mitosis. CDK1-Cyclin B further promotes the activation of GWL kinase, which subsequently phosphorylates ENSA and Arpp19. The phosphorylated forms of these endosulfines bind to PP2A-B55 and inhibit the phosphatase activity towards CDK1-Cyclin B substrates, including MYT1, WEE1, and CDC25. (**B**) The CDC25B and CDC25C isoforms become active during late G2 to drive CDK1-Cyclin B activation. During interphase, CDC25B and CDC25C are kept inactive by phosphorylation and are retained in the cytoplasm due to binding to 14-3-3 proteins. The activity of CDC25B is proposed to be responsible for the initial activation of CDK1-Cyclin B at centrosomes. Activation of CDC25B is triggered by Aurora A- and PLK1-dependent phosphorylations on the phosphatase N-terminus, which also promote CDC25B shuttling between the cytoplasm and the nucleus. Following activation, the centrosomal pool of CDK1-Cyclin B is translocated to the nucleus, where it enhances CDC25C activity, initiating an amplification loop that drives the cell into mitosis. The inhibitory phosphorylation on CDC25C is removed by PP1 phosphatase upon dissociation of the 14-3-3 protein. This concurs with PLK1-mediated phosphorylation of the CDC25C N-terminus to promote the phosphatase nuclear import and directly stimulate the phosphatase catalytic activity.

**Figure 4 biomolecules-09-00055-f004:**
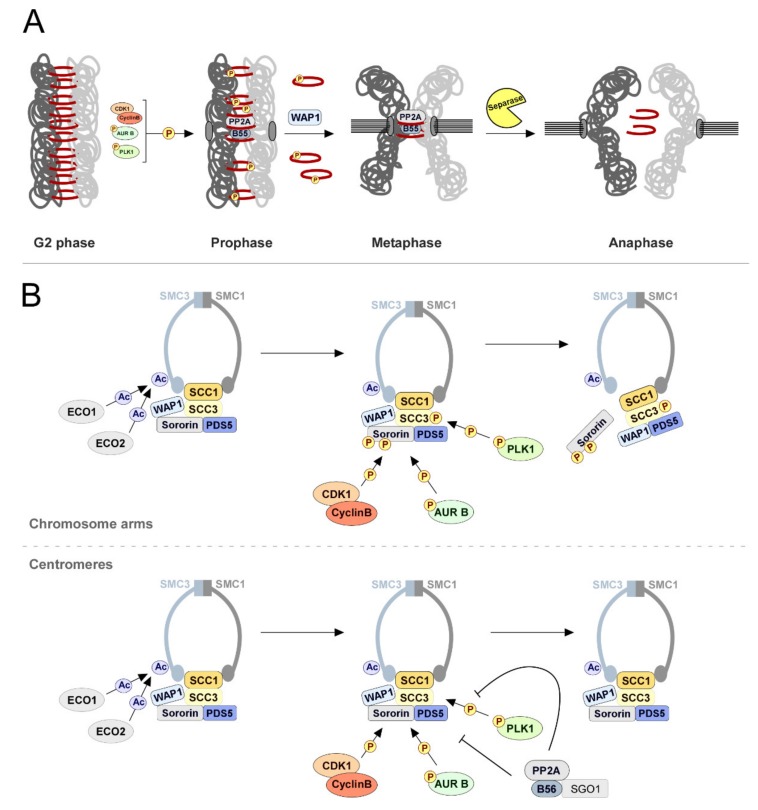
PP2A-B56 maintains centromeric cohesion until anaphase onset. (**A**) Following DNA replication, sister chromatids are held together by cohesin-ring complexes. During prophase, CDK1, PLK1, and Aurora B trigger WAP1-mediated removal of cohesin from chromosome arms. Centromere cohesion is protected from WAP1 by the activity of PP2-B56 bound to SGO1. Centromeric PP2-B56 antagonizes the phosphorylations catalyzed by the mitotic kinases, hence maintaining cohesion at centromeres until kinetochores of all chromosomes are correctly attached to microtubules of opposite spindle poles. At this stage, Separase catalyzes the proteolytic degradation of cohesin complexes, abolishing cohesion and enabling sister chromatid separation in anaphase. (**B**) During G2, cohesin rings are insensitive to WAP1 by Sororin recruited to acetylated SMC3. The presence of Sororin prevents WAPl binding to PDS5. Following mitotic entry, Aurora B- and CDK1-dependent phosphorylation of Sororin causes its dissociation, thus consenting WAP1 binding to PDS5. WAP1-mediated removal of cohesin also requires phosphorylation of SCC3 by PLK1. At centromeres, PP2A-B56 in complex with SGO1 counteracts Aurora B, CDK1, and PLK1 destabilizing phosphorylations and thereby prevents the dissociation of cohesin complexes by WAP1.

**Figure 5 biomolecules-09-00055-f005:**
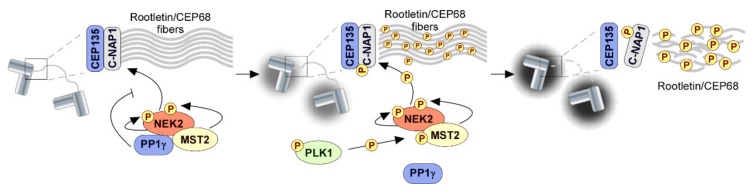
PP1 prevents premature separation of centrosomes. Centrosome splitting during late G2 involves the dissolution of the Rootletin/CEP68 linker that holds duplicated centrosomes together. This process is orchestrated by NEK2A-mediated phosphorylation of C-Nap1 and of linker proteins. Phosphorylation of C-NAP1 abrogates its interaction with CEP135 docked on the proximal ends of parental centrioles, thus causing C-NAP1 displacement from centrosomes. Phosphorylation of Rootletin and of associated proteins disrupts the linker structure and enables centrosome disjunction. PP1γ suppresses NEK2A activity up until late G2, thereby preventing premature resolution of centrosomes. During interphase, PP1γ bound to NEK2A in complex with MST2 is able to efficiently antagonize phosphorylations catalyzed by the kinase. Increasing activity of PLK1 in late G2 results in MST2 phosphorylation and consequential dissociation of PP1γ from the complex. This renders NEK2A phosphorylations unopposed, hence allowing pre-mitotic resolution of centrosomes.

**Figure 6 biomolecules-09-00055-f006:**
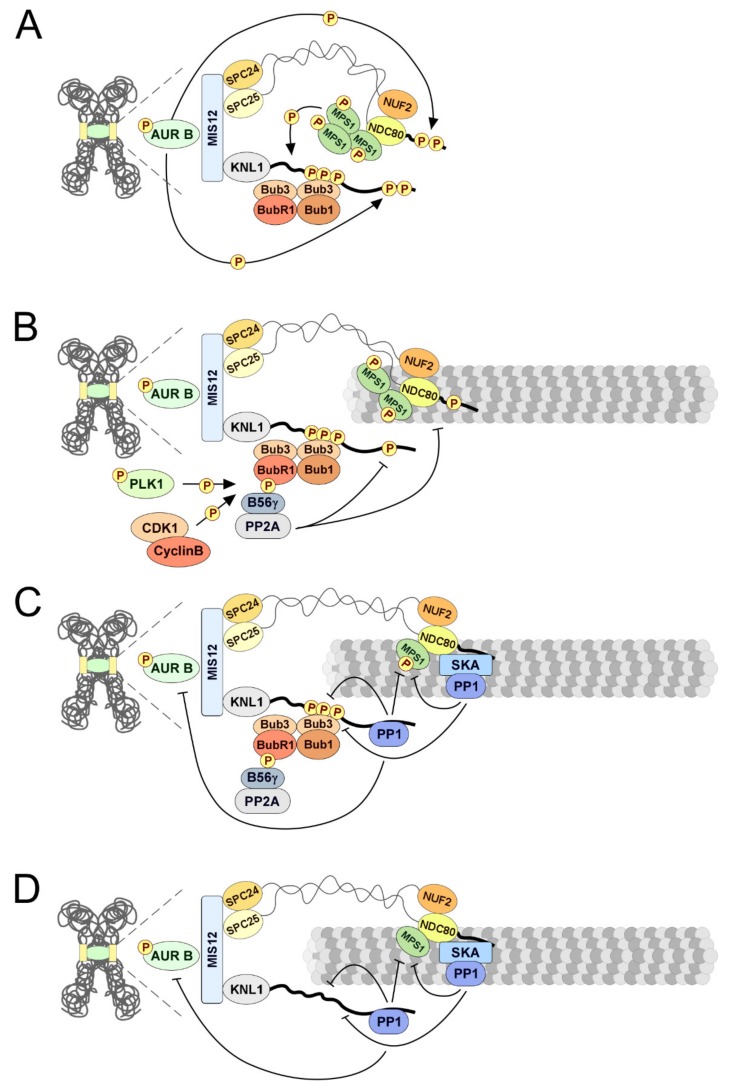
Kinetochore PP2A-B56 and PP1 promote the stability of microtubule attachments and efficient Spindle Assembly Checkpoint silencing in animal cells. (**A**,**B**) In the absence of microtubules, MPS1 kinase efficiently accumulates at kinetochores through direct binding to the NDC80 complex. This enables MPS1 trans-autoactivation and instates SAC signaling. MPS1 phosphorylates several MELT motifs on KNL1, generating docking sites for the hierarchical recruitment of SAC proteins required for MCC assembly. This also enables the recruitment of PP2A-B56γ to kinetochores, following PLK1-mediated phosphorylation of BUBR1 KARD motif. PP2A-B56γ bound to BUBR1 has critical roles in allowing the initial binding of microtubules to NDC80 and in establishing SAC responsiveness. The activity of PP2A-B56γ-BUBR1 antagonizes microtubule-destabilizing phosphorylations on the NDC80 N-terminal tail. These phosphorylations are catalyzed by centromeric Aurora B and predominate at tensionless kinetochores to enable the correction of erroneous attachments. Dephosphorylation of the NDC80 N-terminal tail by PP2A-B56-BUBR1 provides NDC80 an opportunity to bind microtubules. (**C**,**D**) As microtubules occupancy increases, kinetochore tension or alterations on the kinetochore architecture place outer-kinetochore proteins further apart from the Aurora B zone of influence. This renders PP2A-B56γ-BUBR1 activity largely unopposed, which ultimately results in complete dephosphorylation of NDC80 and in the formation of stable end-on attachments. At this point, several pools of PP1γ accumulate at the kinetochore to preserve the stability of bioriented attachments and promote prompt SAC silencing. The recruitment of PP1γ is mediated in part by its interaction with PP1-binding motifs on the KNL1 N-terminus. PP1γ-KNL1 dephosphorylates phospho-MELTs, causing the dissociation of SAC proteins from kinetochores and consequently ceasing MCC assembly. Binding of PP1γ to KNL1 is limited at unattached or tensionless kinetochores due to Aurora B-mediated phosphorylation of both PP1-docking motifs, which significantly represses their ability to interact with the phosphatase. These phosphorylations are counteracted by the activity of PP2A-B56γ bound to BUBR1. Thus, as end-on attachments are established, the KNL1 N-terminus becomes inaccessible to Aurora B and PP2A-B56γ ensures efficient recruitment of PP1γ. Increasing levels of PP1γ-KNL1 further suppress Aurora B activity at kinetochores, hence enhancing the stability of microtubule attachments. PP1γ-KNL1 is also proposed to directly antagonize the activating T-loop autophosphorylation of Aurora B, which likely favors the recruitment of the SKA complex to dephosphorylated NDC80. The SKA complex directly binds PP1γ through its SKA1 subunit and independently of KNL1, thus representing an additional pathway to deliver PP1γ to kinetochores of bioriented chromosomes. Importantly, because MPS1 and microtubules have partially overlapping binding surfaces on the NDC80 complex, the formation of stable end-on attachments imposes a substantial reduction in the amount of MPS1 at kinetochores. Residual MPS1 molecules remaining on bioriented kinetochores are dephosphorylated and inactivated by PP1. These mechanisms concur to efficiently avert SAC function at the top of the signaling cascade.

**Figure 7 biomolecules-09-00055-f007:**
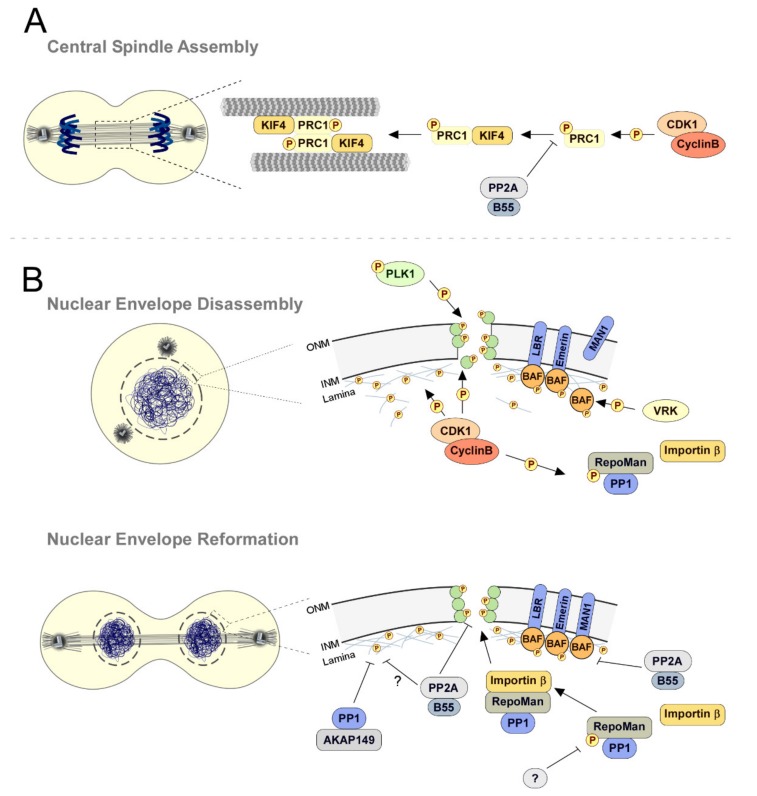
PP2A-B55 and PP1 activities orchestrate mitotic exit. (**A**) Formation of the central spindle during anaphase relies on the activity of PRC1. Phosphorylation of PRC1 by CDK1-Cyclin B precludes its microtubule bundling activity. Dephosphorylation of PRC1 is catalyzed by PP2A-B55 during anaphase and enables PRC1 interaction with the kinesin motor KIF4, its consequent translocation to the plus ends of antiparallel interdigitating microtubules, and the protein homodimerization required for its microtubule bundling activity and thereby for the assembly of the central spindle. (**B**) Reformation of the nuclear envelope and associated nuclear pore complexes around chromatin occurs at the end of mitosis. This requires removal of the multiple phosphorylations that were catalyzed during mitotic entry to promote the disassembly of the nuclear envelope and of nuclear pore complexes. During late anaphase/telophase, PP2A-B55-mediated dephosphorylation of BAF restores its interaction with chromatin and with components of the inner nuclear membrane (INM). PP2A-B55 might also target Lamin and several nucleoporins to direct their assembly into the nuclear envelope. Multiple pools of PP1 cooperate with PP2A-B55 to promote nuclear envelope reformation. AKAP149 localizes PP1 on the assembling nuclear envelope, where it dephosphorylates Lamin B to promote its polymerization and accelerate nuclear lamina assembly. Repo-Man targets PP1γ to the periphery of late anaphase chromosomes. The N-terminus of Repo-Man directly mediates the recruitment of Importin-β to control early nuclear envelope formation and nucleoporin deposition.

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
