# Peer review of "Phosphatases in Mitosis: Roles and Regulation"

_biomolecules, 2019, doi:10.3390/biom9020055_

Reviewer 1 Report

The review by Moura and Conde provides valuable and extensive insights into the role of phosphatases in mitosis. The authors have systematically reported the importance of various phosphatases in controlling mitotic progression in a spatiotemporal manner. They began with describing in details structure and activities of multiple phosphatases (PP1, PP2A, Tyrosine phosphatases, and CDC25 phosphatase). Thereafter they described the function of these phosphatases in controlling various aspects of mitosis namely mitotic entry, chromatid cohesion, spindle assembly, kinetochore microtubules attachments, spindle assembly checkpoint, and mitotic exit and cytokinesis. 

I believe that this review will give a perfect overview and unravel the earlier 'unappreciated' relevance of phosphatases in mitosis, and I fully support this review article. I have only a few minor comments that may be useful to improve this manuscript further.

1. In the context of 'to build a spindle' on p. 12, authors can consider citing [Han et al., (2009), Genetics 181, 933-943] that reveal the role of PP4 in microtubule severing.

2.  Authors can further consider elaborating on the less known function of phosphatases on spindle orientation. For instance, they can cite the work from Knoblich lab [Xie et al., (2013) Neuron 79, 254-265]. Further, they may like to discuss the function of PP1/Repo-Man and PP6/Aurora A cross-talk in spindle orientation [Ho Lee et al., (2017) JCB 217, 483; Kotak et al., (2016) JCS 129, 3015-3025 and Afshar et al., (2010) Development, 137, 237-247]

3. On p.  13 in the sentence 'This is owed....the assembling spindle'   the references 134 and 243 seems wrong. 

4. On p. 24 in the section of spatiotemporal control of phosphatase activities, in addition to emphasizing the function of the polybasic region in the de-phosphorylation during mitotic exit, Moura and Conde can further discuss the preferential role of having Cdk1 sites as Threonine in comparison to serine  for early vs late substrates upon mitotic exit (Refs. 63 and 64). 

5.Authors like to replace the word 'cover' with 'mask' on p.9 line 375

Author Response

The review by Moura and Conde provides valuable and extensive insights into the role of phosphatases in mitosis. The authors have systematically reported the importance of various phosphatases in controlling mitotic progression in a spatiotemporal manner. They began with describing in details structure and activities of multiple phosphatases (PP1, PP2A, Tyrosine phosphatases, and CDC25 phosphatase). Thereafter they described the function of these phosphatases in controlling various aspects of mitosis namely mitotic entry, chromatid cohesion, spindle assembly, kinetochore microtubules attachments, spindle assembly checkpoint, and mitotic exit and cytokinesis.

I believe that this review will give a perfect overview and unravel the earlier 'unappreciated' relevance of phosphatases in mitosis, and I fully support this review article. I have only a few minor comments that may be useful to improve this manuscript further.

We thank the reviewer for the critical and constructive evaluation of the manuscript. We are pleased that the reviewer recognizes our effort in providing a thorough overview on the function and regulation of mitotic phosphatases and that this represents a valuable document for the cell cycle field. We found the reviewer comments and suggestions very useful and addressed them accordingly.         

Point 1: In the context of 'to build a spindle' on p. 12, authors can consider citing [Han et al., (2009), Genetics 181, 933-943] that reveal the role of PP4 in microtubule severing.

Response 1: We followed the reviewer suggestion and included this information in the revised version of the manuscript (line 524). 

Point 2: Authors can further consider elaborating on the less known function of phosphatases on spindle orientation. For instance, they can cite the work from Knoblich lab [Xie et al., (2013) Neuron 79, 254-265]. Further, they may like to discuss the function of PP1/Repo-Man and PP6/Aurora A cross-talk in spindle orientation [Ho Lee et al., (2017) JCB 217, 483; Kotak et al., (2016) JCS 129, 3015-3025 and Afshar et al., (2010) Development, 137, 237-247]

Response 2: We agree with the reviewer and included a new paragraph in the revised version of the manuscript describing the involvement of mitotic phosphatases in the control of spindle orientation (line 714).    

Point 3: On p.  13 in the sentence 'This is owed....the assembling spindle'   the references 134 and 243 seems wrong

Response 3: The reviewer is correct. We apologize for the error and removed the references accordingly.  

Point 4: On p. 24 in the section of spatiotemporal control of phosphatase activities, in addition to emphasizing the function of the polybasic region in the de-phosphorylation during mitotic exit, Moura and Conde can further discuss the preferential role of having Cdk1 sites as Threonine in comparison to serine for early vs late substrates upon mitotic exit (Refs. 63 and 64).

Response 4: We followed the reviewer suggestion and accordingly discuss in the revised version of the manuscript the phosphothreonine preference of PP2A as an additional element controlling the order of mitotic exit events (line 1729).     

Point 5: Authors like to replace the word 'cover' with 'mask' on p.9 line 375

Response 5: We edited in the text accordingly.

Reviewer 2 Report

The manuscript describing the regulatory role of protein phosphatases in cell cycle control provides important insights into cell cycle progression. The comparison between different phosphatases have been well highlighted in the text as well as through illustrations. 

It is ready for publication without any further changes.   

Author Response

The manuscript describing the regulatory role of protein phosphatases in cell cycle control provides important insights into cell cycle progression. The comparison between different phosphatases have been well highlighted in the text as well as through illustrations.

It is ready for publication without any further changes.

We thank the reviewer for valuing our work.